# Functional imaging of conduction dynamics in cortical and spinal axons

**Milos Radivojevic\*, Anna Rostedt Punga**

Uppsala University, Department of Medical Sciences, Uppsala, Sweden

**Abstract** Mammalian axons are specialized for transmitting action potentials to targets within the central and peripheral nervous system. A growing body of evidence suggests that, besides signal conduction, axons play essential roles in neural information processing, and their malfunctions are common hallmarks of neurodegenerative diseases. The technologies available to study axonal function and structure integrally limit the comprehension of axon neurobiology. High-density micro-electrode arrays (HD-MEAs) allow for accessing axonal action potentials at high spatiotemporal resolution, but provide no insights on axonal morphology. Here, we demonstrate a method for electrical visualization of axonal morphologies based on extracellular action potentials recorded from cortical and motor neurons using HD-MEAs. The method enabled us to reconstruct up to 5-cm-long axonal arbors and directly monitor axonal conduction across thousands of recording sites. We reconstructed 1.86 m of cortical and spinal axons in total and found specific features in their structure and function.

## Editor's evaluation

This work provides fundamental new insight into fine axonal morphologies based solely on extracellular action potential recordings. They provide compelling evidence of fine resolution in mapping functional connections between neurons. The work may have broad use in neurobiology, bioengineering, stem cell biology, as well as tissue engineering in functional characterization.

**\*For correspondence:**
rmilosh@gmail.com

**Competing interest:** The authors declare that no competing interests exist.

## Introduction

Axons are neuronal processes specialized for the conduction of action potentials (APs). Cortical axons serve as communication cables between various types of neurons that are synaptically connected and, accordingly, arranged in multiple layers to receive, process, and convey neural information between different regions in the brain. Motor neurons located in the ventral horn of the spinal cord project their axons outside the central nervous system (CNS) and are responsible for the contraction of effector muscles in the periphery. Spinal axons are specialized to innervate and precisely control different types of muscle fibers, thus ensuring refined coordination of complex body movements (*Stifani, 2014*).

Mainly due to difficulties to experimentally access axonal conduction, axonal information processing has been neglected, and axons are classically seen as conductive cables that do nothing more than faithfully transmit APs in an all-or-none, 'binary' fashion (*Hodgkin and Huxley, 1952*). Later studies have challenged this view and suggested that axons have much more complex roles than traditionally thought (*Alcami and El Hady, 2019*). Contrary to classical concepts, reports show that, besides conducting binary APs, hippocampal and cortical axons can also transmit analog currents in a passive fashion (*Shu et al., 2006*; *Alle and Geiger, 2008*). It has been shown that analog currents integrate with ongoing axonal APs, changing their waveforms and, consequently, affecting synaptic transmission in a graded, 'analog' manner (*Shu et al., 2006*; *Alle and Geiger, 2006*; *Zbili and Debanne, 2019*). Additionally, it was found that axons modulate the waveforms of APs contingent on neuronal activity and, accordingly, adjust the synaptic release (*Geiger and Jonas, 2000*). Moreover, during

high-frequency regimes of neuronal activity, axons are able to reduce their conduction velocities by more than 20% and, thereby, tune the timing of AP arrival at synapses (*Radivojevic et al., 2017*). Taken together, these findings suggest that axons passively and actively process APs to tune the amount of information transmitted by synapses, but also imply that axons play a crucial role in the temporal coding of the neural information.

Morphological complexity of mammalian axons generally depends on their downstream targets' spatial disposition. Thus, for example, cortical neurons form extensively branched axons to convey APs to numerous postsynaptic neurons located in different cortical layers and regions in the brain. Due to their small diameter of 0.08–0.4 μm (*Debanne, 2004*) and absence of surrounding myelin sheaths, cortical axons provide relatively low conduction velocities of 0.1–2 m/s (*Radivojevic et al., 2017*; *Bakkum et al., 2013*), which are, nevertheless, sufficient to ensure rapid communication between closely spaced neurons. As opposed to cortical neurons, motor neurons transmit signals to distant targets outside the CNS and, for that purpose, develop considerably longer (up to 1 m), thicker (0.5–10 μm), and myelinated axons that provide considerably higher conduction velocities of up to 100 m/s (*Debanne et al., 2011*; *Saliani et al., 2017*).

Despite being acknowledged as reliable conduction cables, axons can in certain cases fail to propagate APs faithfully. Such cases are referred to as conduction failures and are attributed to particular axonal morphology. For instance, experimental and theoretical studies suggest that conduction failures are likely to occur at axonal branching points and local swellings due to abruptly increased axon diameter (*Debanne, 2004*). In addition, the AP propagation fidelity and temporal precision depend on axon diameter, biophysical properties of various types of ion channels, and thermodynamic noise inherent to their gating dynamics (*Faisal and Laughlin, 2007*; *Faisal et al., 2005*; *White et al., 2000*). According to theoretical studies, thin distal axons (diameter <1 μm) are prone to 'channel noise', which can introduce variability in axonal AP waveforms (*Neishabouri and Faisal, 2014*), increase the jitter of AP propagation (*Radivojevic et al., 2017*; *Faisal and Laughlin, 2007*), and compromise the reliability of the conduction itself (*Faisal and Laughlin, 2007*; *Skaugen and Walløe, 1979*; *Schneidman et al., 1998*). Besides geometrical factors, neuronal activity can also affect AP conduction, which is consistent with our previous finding that high-frequency neuronal activity increases the jitter of AP propagation in cortical axons (*Radivojevic et al., 2017*).

Ultrastructural aberrations and malfunctioning conduction in human axons are often early hallmarks of neurodegenerative diseases. For instance, in Alzheimer's disease (AD), abnormal protein aggregates cause local swellings in axons which have been shown to reduce conduction velocity and even block AP propagation (*Blazquez-Llorca et al., 2017*). Human post-mortem studies suggest that axonal degeneration may be the earliest feature of Parkinson's disease and, therefore, an appropriate target for early therapeutic interventions (*Burke and O'Malley, 2013*). Motor neuron degeneration is the hallmark of amyotrophic lateral sclerosis (ALS), where axonal dysfunction begins long before symptom onset and motor neuron death (*Suzuki et al., 2020*).

Comprehension of axon neurobiology in health and disease is generally limited by the technologies available to study axonal function and structure integrally. The whole-cell patch-clamp technique, complemented with fluorescence microscopy, is commonly used to correlate electrophysiological data with the morphological properties of the neuron. Namely, a patch-clamp pipette can inject fluorescent dyes directly into the neuron and, therefore, allow visualization of the axonal arbor during the electrophysiological experiment. However, patch-clamp is typically limited to recording APs from a single axonal site (*Shu et al., 2006*; *Alle and Geiger, 2006*; *Forsythe, 1994*) and does not allow for tracking AP propagation across axons. Moreover, the technique is invasive and destructive, which constrains the duration of a recording session to about an hour. Alternatively, fluorescent indicators sensitive to voltage (*Peterka et al., 2011*) or calcium (*Grienberger and Konnerth, 2012*) can be used to observe axonal AP propagation and, at the same time, to visualize axonal morphology. However, fluorescent indicators exhibit photobleaching and phototoxicity and may perturb the physiology of the cell to the point of affecting AP conduction (*Peterka et al., 2011*). Complementary-metal-oxide-semiconductor (CMOS)-based high-density microelectrode arrays (HD-MEAs) have been designed to record extracellular APs from neuronal cultures (*Müller et al., 2015*) and allow tracking axonal signals across hundreds of microelectrodes (*Radivojevic et al., 2017*; *Bakkum et al., 2013*). Thanks to a low-noise CMOS design, HD-MEAs enable detection of APs across entire arbors of cortical axons, including tiny axon terminals (*Radivojevic et al., 2017*). HD-MEAs provide noninvasive access to axonal APs and

impose no constraints on the duration of the recording sessions. Nevertheless, HD-MEA technology does not provide direct insights into axonal morphology. It has to be complemented with auxiliary optical techniques to allow correlation between axonal function and structure. Live-cell imaging techniques can be used to optically visualize axonal morphologies directly on HD-MEA surfaces (*Radivojevic et al., 2017*; *Bakkum et al., 2013*; *Bullmann et al., 2019*). These techniques, however, entail introduction of fluorescent reporters into the cell, which induces chemotoxicity, phototoxicity, and cell death (*Icha et al., 2017*).

Our objective was to enable the reconstruction of axonal morphologies based solely on extracellular APs recorded from in vitro mammalian neurons using the CMOS-based HD-MEA system. The present study reports a method for tracking AP propagation across tens-of-millimeter-long nonmyelinated axonal arbors of primary cortical and motor neurons cultured on HD-MEAs. The method allows for label-free electrical visualization of axonal conduction trajectories and, at the same time, provides noninvasive access to axonal APs waveforms recorded across hundreds of microelectrodes. Using the developed method, we investigated (I) morphological features of cortical and spinal axons, (II) fluctuations in AP amplitudes across axonal arbors, and (III) temporal dynamics of the axonal conduction.

## Results

We developed a method for the automatic reconstruction of axonal morphology based on extracellular APs recorded from cortical and motor neurons using the HD-MEA system (*Figure 1A*). The reconstructed morphology is here referred to as 'functional morphology', and it reveals the location of the axon initial segment (AIS), axonal trunk, and higher-order branches (*Figure 1B*). The method was used to trace extracellular APs propagating across axonal arbors of cortical (*Figure 1—video 1*) and motor neurons (*Figure 1—video 2*). The experimental design and overview of the method are outlined in *Figure 2* and are described below.

### Experimental design and overview of the method

We cultured rat primary cortical (*Figure 2—figure supplement 1*) and motor neurons (*Figure 2— figure supplement 2*) on a CMOS-based HD-MEA system that comprises 26,400 densely packed microelectrodes (*Figure 2—figure supplement 3*). The cultures were grown in monolayers with estimated thicknesses of ~5–40 and ~5–50 μm for cortical and motor neuron cultures, respectively (*Figure 2—figure supplement 4*). Both cortical and motor neuron cultures yielded cell densities of ~500–2000 neurons per mm$^2$ (*Figure 2—figure supplement 4*). The cultures matured in controlled conditions (see Methods), and their extracellular electrical activity was recorded between 12 and 24 days in vitro (DIV). Mature neurons grew their axonal arbors efficiently across sensing areas of the HD-MEA chips and provided a tight interface between axons and microelectrodes (*Figure 2—figure supplement 5*). The relative proximity of axons to the sensing area varied across axonal arbors within ranges of ~1–13 and ~1–18 μm for cortical and motor neurons, respectively (*Figure 2—figure supplement 5*).

The CMOS-based HD-MEA system enabled us to map extracellular APs across entire cultures and to electrically identify individual neurons (*Figure 2—figure supplement 6*). Spontaneous neuronal activities were recorded across all microelectrodes for 2 min, and average amplitudes of the recorded voltage traces were used to produce network-wide activity maps (*Figure 2—figure supplement 6A*). Because extracellular APs with the largest amplitudes arise from the AIS (*Bakkum et al., 2019*), local maxima found within these maps indicated the location of individual neurons in the network (*Radivojevic et al., 2016*).

We used spike-sorting algorithms to discern signals among individual neurons in the culture (see Methods). The dense arrangement of the microelectrodes allowed us to access the electrical activity of a single neuron at high spatial resolution (*Figure 2—figure supplement 6A*, *Figure 2—videos 1 and 2*); however, overlapping signals recorded from multiple neighboring neurons were observed in most of the cases (*Figure 2—figure supplement 6B*). Spike-sorting procedures enabled us to discern APs among individual neurons reliably and extract relative times of their activities ('spike times') (*Radivojevic et al., 2017*; *Radivojevic et al., 2016*; *Jäckel et al., 2012*). Sorted APs were then averaged over adjacent electrodes to reveal the spatiotemporal distribution of a single neuron's activity. The spatiotemporal distribution of APs recorded from proximal neuronal compartments (near the AIS) is

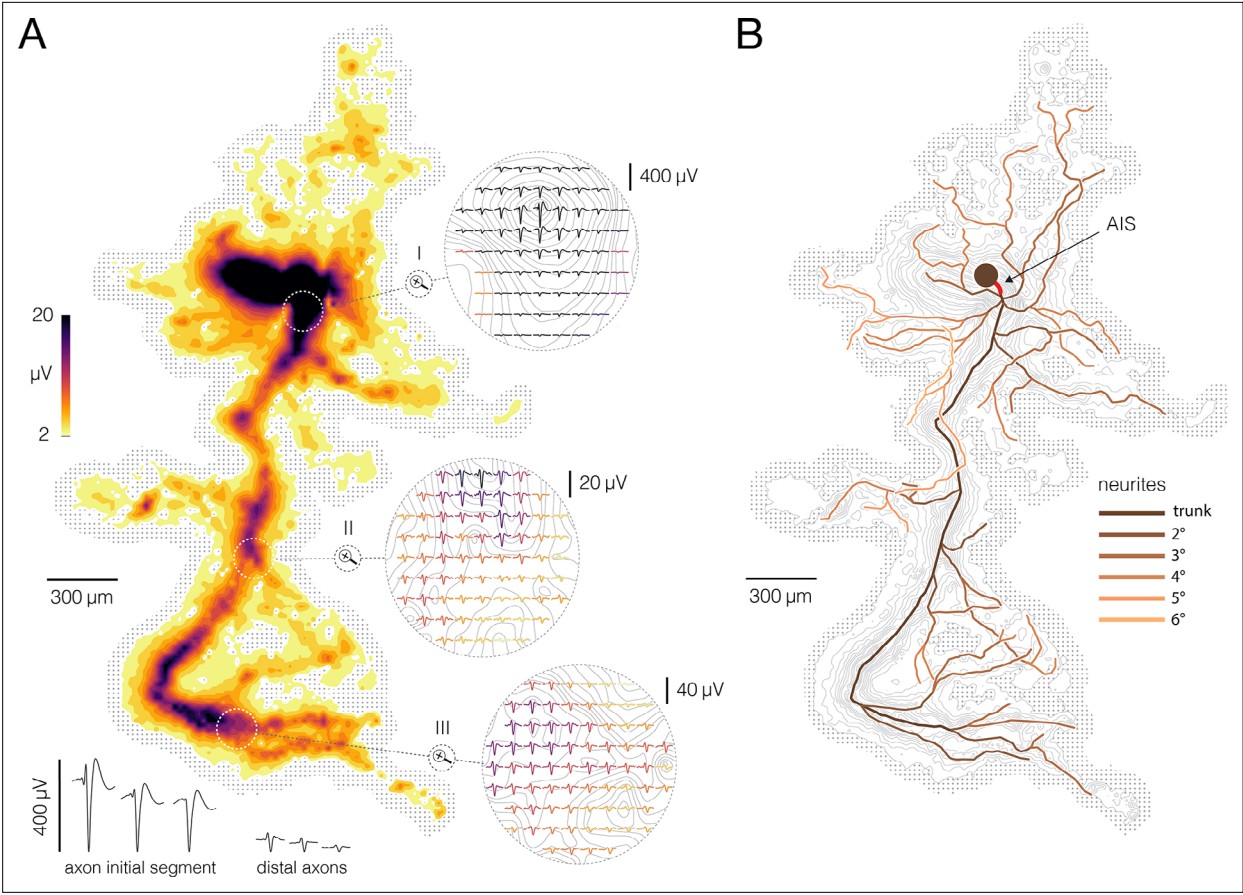

**Figure 1.** Reconstruction of axonal morphology based on neuronal electrical activity. (**A**) Contour map shows spatial distribution of extracellular action potentials (APs) recorded from a single cortical neuron. Average AP amplitudes are color-coded and presented through contour surfaces. Gray dots in the background represent locations of the recording electrodes. White-dashed-line circles superimposed over the contour map denote three magnified regions displayed on the right. Average AP waveforms obtained from proximal, middle, and distal axons are shown in the three denoted regions (labeled I, II, and III, respectively); color-coding is the same as in the contour map. Examples of AP waveforms recorded from the axon initial segment (AIS) and distal axons are shown at the bottom. (**B**) Functional morphology of axonal arbor reconstructed from APs displayed in (**A**). Branching orders of reconstructed neurites are color-coded. The AIS is depicted by a thick red line superimposed over the most proximal part of the axon. A filled dark-brown circle represents putative location of the neuronal soma. Color-free contour map and recording electrodes are shown in the background (same as in A). Functional morphologies of cortical and spinal axons are also presented in *Figure 1—videos 1 and 2*, respectively.

The online version of this article includes the following video(s) for figure 1:

**Figure 1—video 1.** Functional morphologies of cortical axons displayed in real time.

https://elifesciences.org/articles/86512/figures#fig1video1

**Figure 1—video 2.** Functional morphologies of spinal axons displayed in real time.

https://elifesciences.org/articles/86512/figures#fig1video2

called an 'extracellular AP footprint'. Electrical footprints reconstructed for eight neighboring neurons are presented in *Figure 2—figure supplement 6C*. Z-stack image series of the corresponding culture is shown in *Figure 2—videos 1 and 2*.

We used spike-triggered averaging for electrical imaging of axonal arbors (*Figure 2—figure supplement 7*). High-amplitude APs detected near the AIS were used to trigger the averaging of single voltage traces ('single trials') recorded across all electrodes in the array (*Figure 2—figure supplement 7A*). Because the AIS signal represents the first occurring (initial) trace of the neuron's activity, the averaging reveals spatial and temporal shifts in propagating axonal signals (*Figure 2—figure supplement 7B*). The spatiotemporal distribution of axonal signals reconstructed using spike-trigger averaging is called an 'axonal electrical image'. Axonal electrical images of three neighboring neurons are presented in *Figure 2—figure supplement 7B* and in *Figure 2—video 3*.

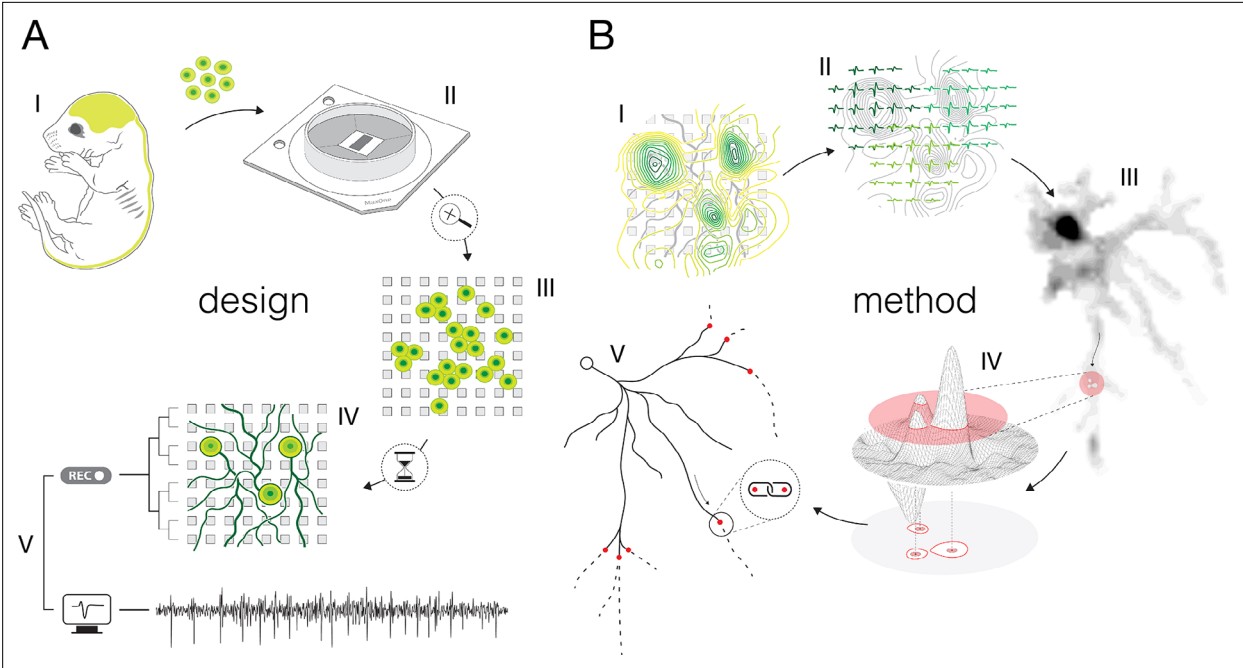

**Figure 2.** Experimental design and overview of the method. (**A**) Experimental design: (**I**) Brain cortices and spinal cords isolated (lime-green) from Sprague Dawley rat embryos were used as sources of primary cortical and motor neurons. Cells obtained from corresponding sources were seeded directly on high-density microelectrode arrays (HD-MEAs) to grow cortical and motor neurons in separate cultures. (**II**) The utilized HD-MEA system features 26,400 microelectrodes packed within an area of 3.85×2.1 mm². Interelectrode distances of 17.5 μm provide density of 3264 microelectrodes per mm². (**III**) The HD-MEA surface coated with cell-adhesive materials promoted neuronal growth over the sensing area and provided efficient electrode-to-neuron contacts. (**IV**) Dedicated media and growth-promoting factors were used to develop and maintain the cultures in vitro over extended periods of time. (**V**) Mature cultures conveyed spontaneous electrical activities. Neuronal extracellular action potentials (APs) were recorded through HD-MEA recording channels at 20 kHz sampling rate. (**B**) Overview of the method: (**I**) Signal amplitudes sampled across an entire HD-MEA were used to produce network-wide activity maps (yellow-green contour map). Local peaks found in activity maps indicated locations of individual neurons in the network. (**II**) Spike-sorting of recorded signals enabled to discern activities between neighboring neurons and to extract their individual electrical profiles. Displayed electrical profiles (dark-, mint-, and lime-green waveforms) are referred to as 'extracellular AP footprints'. (**III**) Array-wide spike-triggered averaging revealed spatiotemporal distribution of APs across axons of individual neurons (gray-scale signal map). Such representation of neuronal signals is referred to as an 'axonal electrical image'. (**IV**) Dynamic thresholding was used to detect local peaks within the footprint. (**V**) Detected peaks were interlinked based on their spatial and temporal proximities to reconstruct axonal conduction trajectories.

The online version of this article includes the following video and figure supplement(s) for figure 2:

**Figure supplement 1.** Complementary-metal-oxide-semiconductor (CMOS)-based high-density microelectrode array (HD-MEA).

**Figure supplement 2.** Culturing primary cortical neurons on the high-density microelectrode array (HD-MEA).

**Figure supplement 3.** Culturing primary motor neurons on the high-density microelectrode array (HD-MEA).

**Figure supplement 4.** Thickness and cell density of neuronal cultures grown on the high-density microelectrode array (HD-MEA).

**Figure supplement 5.** Disposition of cortical and spinal neurites with respect to the high-density microelectrode array (HD-MEA) surface.

**Figure supplement 6.** Electrical identification of individual neurons in the culture.

**Figure supplement 7.** Electrical imaging of axonal arbors.

**Figure 2—video 1.** Cortical culture grown on high-density microelectrode array (HD-MEA) surface – ×20 magnification.
https://elifesciences.org/articles/86512/figures#fig2video1

**Figure 2—video 2.** Cortical culture grown on high-density microelectrode array (HD-MEA) surface – ×40 magnification.
https://elifesciences.org/articles/86512/figures#fig2video2

**Figure 2—video 3.** Electrical imaging of axonal arbors.
https://elifesciences.org/articles/86512/figures#fig2video3

We developed a method for reconstructing axonal functional morphologies based purely on features extracted from axonal electrical images. Adaptive thresholding was used to map signal peaks across axonal arbors (*Figure 3*), and the moving object tracking technique was used to reconstruct axonal conduction trajectories (*Figure 4*). The method for reconstruction of axonal functional morphologies is described in the following two sections.

## Adaptive thresholding enables mapping of axonal signal peaks in space and time

Adaptive thresholding applied to axonal electrical images enabled us to map AP peaks propagating across axonal arbors (*Figure 3*). Axonal electrical images were obtained by averaging 200 voltage traces per electrode. Time derivatives of averaged traces (μV/μs) were computed for each of the electrodes, and the resulting data were divided into 400 consecutive timeframes (with 50 μs inter-frame intervals). Adaptive thresholds were next applied at each timeframe to detect AP peaks at different time points during axonal propagation.

Our thresholding technique generally utilized a 'greedy algorithm' principle (see Discussion). The algorithm allowed for recurrent adaptations of the threshold based on parameters updated along the detection process. High-amplitude APs were mapped first, and parameters obtained from the mapped locations were used to tailor the detection of low-amplitude signals in the later steps. The thresholds were initially determined based on electrical noise observed from the HD-MEA chip. Electrical noise was estimated from voltage traces sampled across an entire array during periods when the observed neuron was inactive, and the noise was estimated for each neuron separately.

Adaptive thresholding involves three steps (*Figure 3*, *Figure 3—videos 1–3*). In the first step, a simple planar threshold, set to 9 STD of the estimated noise, was used to detect high-amplitude signal peaks (*Figure 3A*, *Figure 3—video 1*). A high threshold level allowed us to detect AP peaks far above the background noise. However, axonal APs with lower amplitudes remained undetected, leaving gaps along axonal conduction trajectories. In the second step, confined thresholds, set to 2 STD of the estimated noise, were applied locally to detect low-amplitude AP peaks (*Figure 3B*, *Figure 3—video 2*). The confined thresholds were positioned on spatial and temporal coordinates of previously mapped peaks. The thresholds were confined spatially to 50 μm radii and temporally to periods encompassing three consecutive timeframes ($t_{previous}$, $t_{current}$, $t_{next}$, see *Figure 3B*). The spatio-temporal confinement enabled us to detect low-amplitude signals in close proximity to previously mapped peaks and to fill local gaps along the conduction trajectories. The third step utilized the same detection strategy but with differently tuned parameters (*Figure 3C*, *Figure 3—video 3*). Namely, the threshold level was further lowered to 1 STD of the estimated noise, and the spatial confinement was broadened to a 100 μm radius. These parameters enabled the detection of low-amplitude APs near axon terminals.

## A moving object tracking technique enables reconstruction of axonal conduction trajectories

We developed an algorithm for tracking mapped axonal signals across consecutive timeframes (*Figure 4*). The algorithm enabled us to reveal signal conduction trajectories and reconstruct functional morphologies of cortical (*Figure 1—video 1*) and spinal axons (*Figure 1—video 2*).

Our tracking algorithm was designed to predict axonal conduction trajectories based on three factors: (I) spatiotemporal proximities of the mapped peaks, (II) topology of skeletonized axonal signal, and (III) conduction velocities estimated from previously reconstructed trajectories.

The tracking procedure involved three iterative steps (*Figure 4*, *Figure 4—videos 1–3*). (I) Direct interconnection – the closest signal peaks found within a 100 μm Euclidean distance and mapped in two consecutive timeframes were interconnected via direct links (*Figure 4A*, *Figure 4—video 1*). Direct interconnection revealed fragments of axonal conduction trajectories but failed to reconstruct axonal branching forks in most cases. Reconstructed fragments were used to calculate axonal conduction velocities. Obtained velocities served as criteria to tailor the tracking procedure in the next two steps. (II) Skeletonization-assisted interconnection – consecutive signal peaks that could not be interconnected directly, but were found within a 200 μm Euclidean range, were interconnected via skeletonized remnants of the axonal signal (*Figure 4B*, *Figure 4—video 2*). Signal amplitudes extracted from the corresponding electrical images were averaged over two consecutive timeframes (Δt=100 μs)

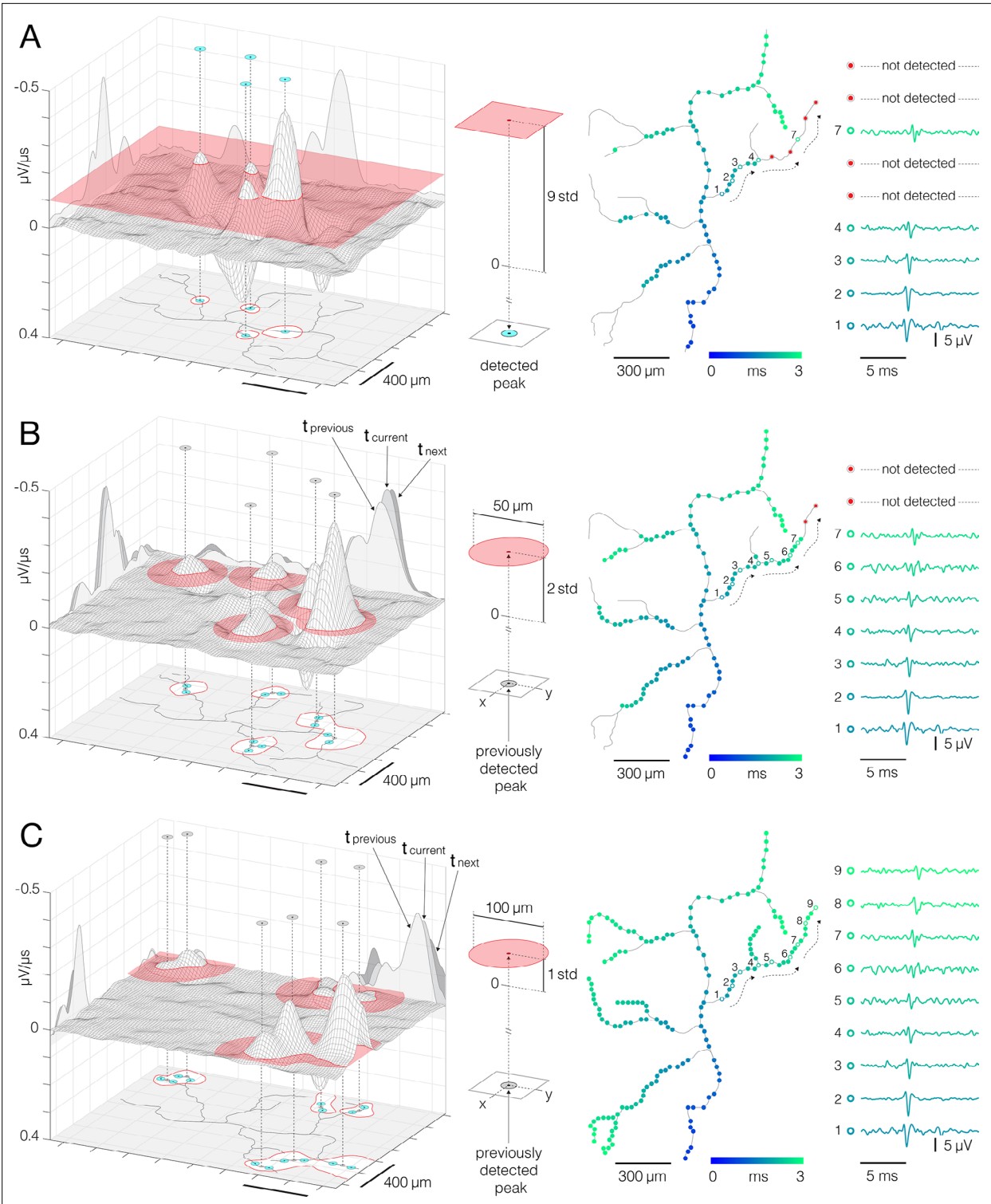

**Figure 3.** Detection of action potentials (APs) propagating across axonal arbors. (**A**) A simple threshold enables detection of high-amplitude signal peaks. (Left) 3D graph shows planar threshold (red semitransparent plane) applied to a section of neuronal signal (3D gray-lined mesh) reconstructed for a single timeframe. 2D profiles of the reconstructed signal are projected on the graph's side planes (gray-filled hills). Signal cutouts found above the threshold are projected on the graph's bottom (red-line bordered white patches). Detected signal peaks are denoted by cyan circles projected perpendicularly from top to the bottom of the graph. Axonal contour shown at the graph's bottom was estimated by observing spatial movements of the signal peaks over consecutive timeframes (also see *Figure 6—videos 1 and 2*). (Middle) Simple planar threshold was set to 9 STD of the estimated noise; background noise was sampled across all electrodes during periods when the neuron was inactive. (Right) Spatiotemporal distribution of the

*Figure 3 continued on next page*

*Figure 3 continued*

detected AP peaks (blue-cyan circles) superimposed over axonal contour (same as in the left panel). Timing of the detected peaks is color-coded. AP waveforms recorded at the numbered locations in the upper-right axonal branch are shown on the side. Black-dashed arrows indicate direction of the propagating APs. Red circles denote signal peaks that were not detected by the simple threshold. (**B, C**) Adaptations of the threshold based on spatial and temporal coordinates of previously detected peaks enabled detection of low-amplitude signals. (Left) Locally applied thresholds (red semitransparent circles) were centered on XY coordinates of previously detected peaks (gray circles projected perpendicularly from top to the bottom of the graph); local thresholds are adapted to detect neighboring peaks at preceding and succeeding timeframes. Neuronal electrical activity reconstructed for a single timeframe (3D gray-lined mesh) is projected on the graph's side planes ($t_{current}$); signal profiles of preceding and succeeding timeframes are also projected ($t_{previous}$, $t_{next}$). Signal cutouts found above the threshold are projected on the graph's bottom (red-line bordered white patches). Newly detected peaks are denoted by cyan circles projected on the graph's bottom plane. Axonal contour same as in (**A**). (Middle) Detection fields of local thresholds are 50 µm (in **B**) and 100 µm (in **C**) in radius, and are set to 2 STD (in **B**) and 1 STD (in **C**) of the estimated noise. (Right) Same as in (**A**); note newly mapped peaks detected near proximal axon. Peak detection strategies are comprehensively demonstrated in *Figure 3—videos 1–3*.

The online version of this article includes the following video(s) for figure 3:

**Figure 3—video 1.** Mapping signals across axonal arbor – step 1.
https://elifesciences.org/articles/86512/figures#fig3video1

**Figure 3—video 2.** Mapping signals across axonal arbor – step 2.
https://elifesciences.org/articles/86512/figures#fig3video2

**Figure 3—video 3.** Mapping signals across axonal arbor – step 3.
https://elifesciences.org/articles/86512/figures#fig3video3

and mapped over interconnection areas. Obtained maps were next skeletonized to infer propagation trajectory between the consecutive signal peaks. Conduction velocities estimated in the previous step were used as criteria for the selection of optimal propagation trajectories. Trajectories whose velocities deviated from previously estimated values by more than 50% were discarded. Skeletonization-assisted interconnection enabled reconstruction of some, but not all, axonal branching forks. It was designed to predict axonal trajectories over consecutive (continuous) signal peaks but could not predict trajectories between discontinuous signal peaks. Axonal conduction velocities estimated from the reconstructed trajectories served as criteria to tailor the tracking procedure in the next step. (III) Indirect interconnection – discontinuous signal peaks mapped in every other timeframe and found within a 400 µm Euclidean range were indirectly interconnected (*Figure 4C*, *Figure 4—video 3*). The propagation trajectory between the discontinuous signal peaks was reconstructed using remnants of skeletonized signals. Signal amplitudes extracted from the corresponding electrical images were averaged over three consecutive timeframes ($\Delta t=150$ µs) and mapped over interconnection areas. Obtained maps were next skeletonized to infer propagation trajectory between signal peaks found in the first and third timeframe. Conduction velocities estimated in the previous steps were used as criteria for selecting optimal propagation trajectories and predicting spatial coordinates of data for the second timeframe. Reconstruction of axonal trajectories over discontinuous signal peaks revealed axonal branching forks that could not be reconstructed in the previous steps.

## Performance of the method for reconstructing axonal functional morphology

The Bayes optimal template-matching technique was used to estimate the algorithm's performance for detecting axonal AP peaks (*Figure 5*). The technique can reliably discriminate axonal APs from the background noise (*Radivojevic et al., 2017*) and therefore provide a ground truth for the validation of the algorithm. Herein, template-matching was used to discriminate between 'false peaks' caused by the noise and 'true peaks' that resulted from axonal electrical activity (*Figure 5A*). Discrimination criteria were based on similarities ('matching') between waveforms of averaged signals ('templates') and corresponding single trials. High similarities were expected in cases of accurate axonal signals (APs) and low similarities in cases where signals were derived from the background noise. We constructed templates for each signal peak in an electrical image. Constructed templates were next compared with waveforms of corresponding single trials, and a percentage of 'matching' trials was computed for each of the peaks. Signal peaks that yielded a match of >70% were classified as true peaks. Classified (true and false) peaks were used as the ground truth to estimate performances of the algorithm for detecting axonal AP peaks (*Figure 5B and C*). Data obtained from 20 cells (10 cortical and 10 motor neurons) were used for this analysis.

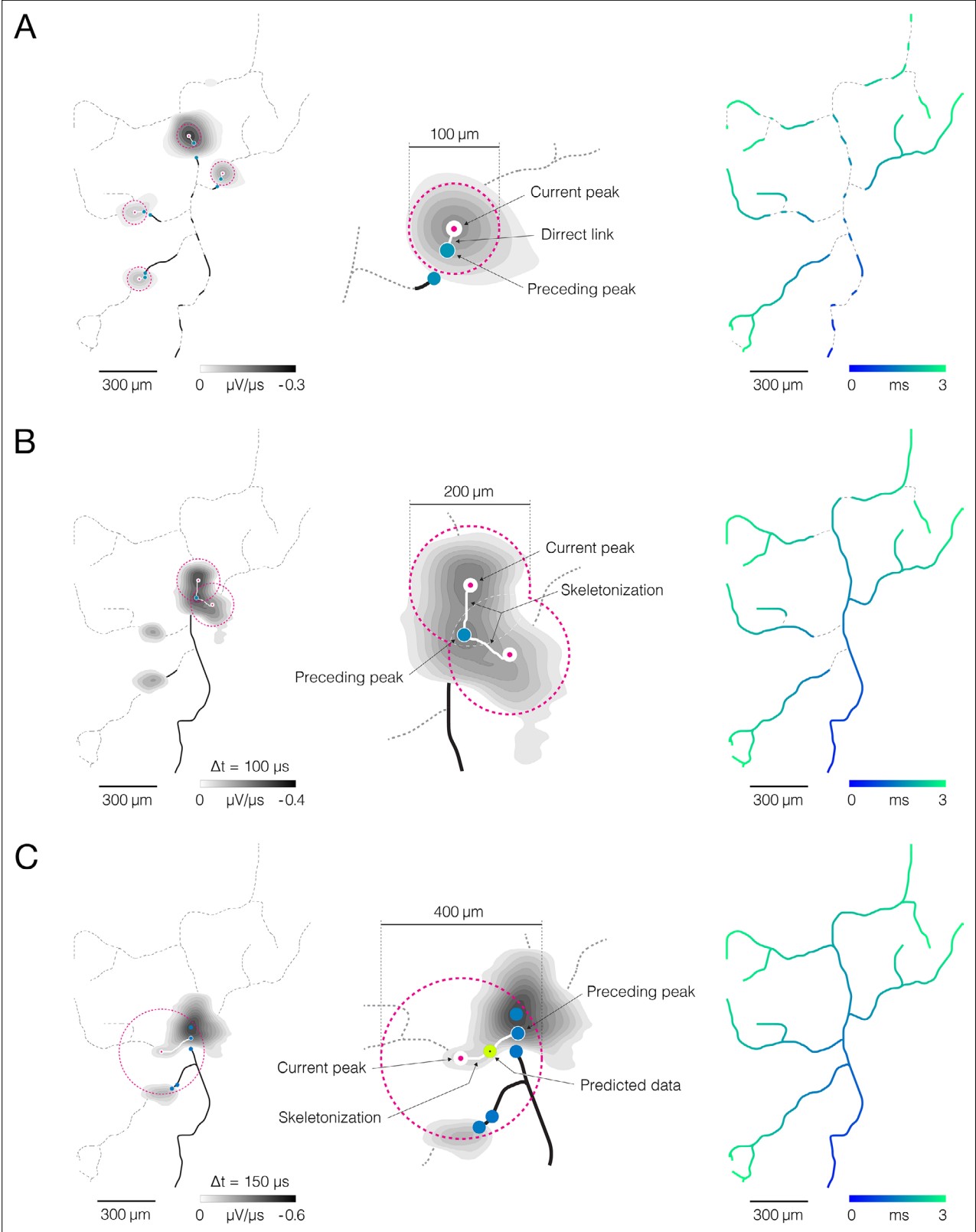

**Figure 4.** Reconstruction of axonal functional morphology. (**A**) Direct interconnection of mapped action potential (AP) peaks. (Left-middle) Closest signal peaks found within a 100 μm Euclidean range (pink-dashed circles) and mapped in two consecutive timeframes (current and preceding peak) were interconnected via direct links (rectilinear white lines). Axonal APs reconstructed for the current timeframe is shown in the background (contour map); average signal amplitude is color-coded (gray scale). (Right) Axonal conduction trajectories revealed by direct interconnection; timing of the

*Figure 4 continued on next page*

*Figure 4 continued*

propagating AP is color-coded (blue-cyan). (**B**) Skeletonization-assisted interconnection of mapped signal peaks. (Left-middle) Signal peaks found within a 200 µm Euclidean range (pink-dashed circles) and mapped at two consecutive timeframes (current and preceding peak) are interconnected via remnants of the skeletonized signal (irregular white lines) – signal averaged over the two consecutive timeframes (Δt=100 µs) is skeletonized to infer directionality of the propagating signal. Axonal signal averaged over the two consecutive timeframes is shown in the background (contour map); signal amplitude is color-coded (gray scale). (Right) Axonal conduction trajectories reconstructed using direct and skeletonization-assisted interconnection; timing of the propagating signal is color-coded (blue-cyan). (**C**) Indirect interconnection of mapped signal peaks. (Left-middle) Signal peaks that were mapped at every other timeframe and found within a 400 µm Euclidean range (pink-dashed circles) were interconnected indirectly – the signal averaged over three consecutive timeframes (Δt=150 µs) was skeletonized to interconnect peaks mapped in first and third timeframe (irregular white line); data for the second timeframe was predicted based on the conduction velocity observed in previously reconstructed trajectories. Axonal signal averaged over the three consecutive timeframes is shown in the background (contour map); signal amplitude is color-coded (gray scale). (Right) Functional axonal morphology reconstructed using direct, skeletonization-assisted and indirect interconnection; timing of the propagating APs is color-coded (blue-cyan). Direct, skeletonization-assisted and indirect interconnections of mapped signal peaks are demonstrated in *Figure 4—videos 1–3*.

The online version of this article includes the following video(s) for figure 4:

**Figure 4—video 1.** Direct interconnection of mapped signal peaks.

https://elifesciences.org/articles/86512/figures#fig4video1

**Figure 4—video 2.** Skeletonization-assisted interconnection of mapped signal peaks.

https://elifesciences.org/articles/86512/figures#fig4video2

**Figure 4—video 3.** Indirect interconnection of mapped signal peaks.

https://elifesciences.org/articles/86512/figures#fig4video3

The performance of the algorithm for the detection of axonal AP peaks is shown in *Figure 5B*. We estimated the performance of the three detection steps (see *Figure 3*) with varying threshold levels. We were able to detect 45%, 74%, and 98% of the actual peaks after the first, second, and third steps, respectively. We observed no false peak detections for thresholds set to 9, 2, and 1 STD of the noise in first, second, and third steps, respectively. The detectability of AP peaks across axonal arbors of cortical and motor neurons is shown in *Figure 5C*.

Axonal conduction trajectories, obtained from stimulation-triggered neuronal activity, were applied to estimate the performance of the algorithm for tracking axonal AP peaks (*Figure 6*). The key concepts of the stimulation protocol used in this study are presented in *Figure 6—figure supplement 1* and described in the Methods section. We used targeted stimulation to reveal and verify the same axon's conduction trajectory by altering its conduction's direction (*Figure 6A*, *Figure 3—video 3*). Stimulation-triggered APs were mapped spatially across all microelectrodes and temporally over discrete timeframes (with 50 µs inter-frame interval) to produce the movie. Observing the spatial movements of AP peaks in consecutive movie frames enabled us to track axonal conduction in different directions visually and to reconstruct axonal conduction trajectories (as shown in *Figure 6A* and *Figure 6—videos 1 and 2*).

The algorithm's performance for tracking axonal conduction is shown in *Figure 6B*. We estimated the performance of the three tracking steps (see *Figure 4*) with varying diameters of corresponding interconnection areas. We could interconnect 70%, 85%, and 91% of the mapped AP peaks after the first, second, and third steps, respectively. We observed no false links when using the interconnection areas with diameters of 100, 200, and 400 µm in first, second, and third steps, respectively. The efficiency of the signal tracking across axonal arbors of cortical and motor neurons is shown in *Figure 6C*.

## Functional morphologies of cortical and spinal axons

The reconstructed functional morphologies revealed axonal arbors of cortical and motor neurons and indicated positions of their AISes and somas (*Figure 7*). They further provided insights into total axonal lengths, the spatial distribution of axonal branching points and terminals, lengths of inter-branching segments and their branching orders. We used electrical images to extract areas occupied by axonal electrical activity ('active areas').

Representative examples of functional morphologies reconstructed for cortical and motor neurons are shown in *Figure 7A*, and displayed in *Figure 7—video 1*. The reconstructed arbor of the cortical axon yielded a total length of 27.12 mm, comprising 101 inter-branching segments and 53 axon terminals. Axonal electrical activity was detected on 7295 electrodes, occupying an active area of 2.23 mm². The axonal arbor of the motor neuron yielded a total length of 15.26 mm, comprising 43

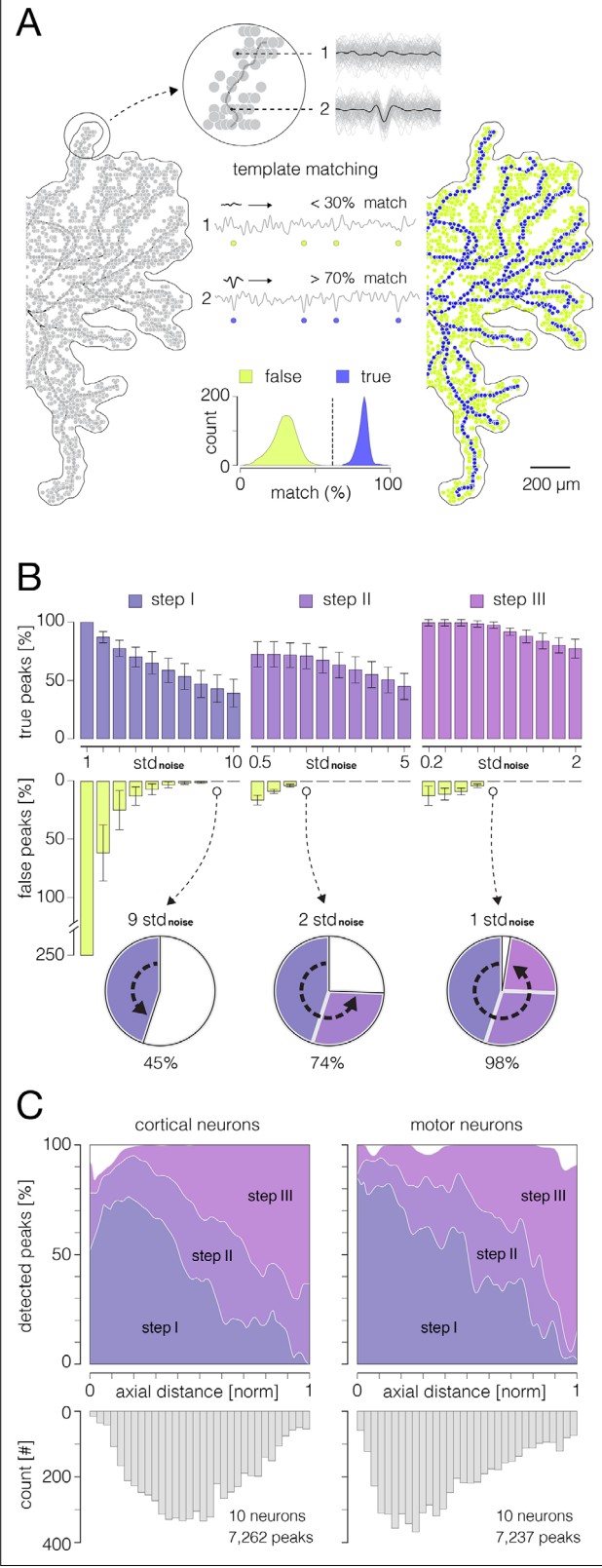

**Figure 5.** Performances of the algorithm for detection of axonal signal peaks. (**A**) Principle of the template-matching demonstrated on data recorded from cortical axons: (Left) Extracellular action potentials (APs) were spike-sorted (see *Figure 2—figure supplement 6*) and reconstructed across an entire array (see *Figure 2— figure supplement 7*) by averaging 100–200 single trials on each electrode. Reconstructed 3D signals were

*Figure 5 continued on next page*

*Figure 5 continued*

next transformed into derivatives (see **Figure 3**) and all available signal peaks were obtained. Obtained peaks (pale-gray circles) are superimposed over the contour of axonal morphology. (Middle) Voltage traces recorded at locations of two example peaks are shown at the top (denoted as 1 and 2); averaged templates (black traces) are superimposed over single recording trials (light-gray traces). Note weak similarities in waveforms between the template and single trials at location 1 (putative noise) as oppose to strong similarities at location 2 (putative axonal signal). Template-matching procedure: the waveform of the template is successively compared with waveforms of discrete recording trials to extract percentage of 'matching' trials within total number of trials. Examples of recording trials are presented continually within voltage traces (denoted as 1 and 2) and their relative positions are marked by circles: yellow and blue circles denote template mismatches and matches, respectively. Histogram below expresses separability between true and false peaks obtained from the template-matching; all obtained peaks (as shown in left) were used for this analysis. Data that yielded a match of >70% were considered true axonal signal. (Right) True and false peaks (blue and yellow circles, respectively) were classified using the template-matching technique. (**B**) Performance of the algorithm for AP peak detection. Histograms show percentages of true and false peak detections across the three thresholding steps (see **Figure 3**) with varying threshold levels. Percentages of true peak detections are exposed through violet, purple, and magenta ascending bars (steps I, II, and III, respectively); percentages of false peak detections are exposed through yellow descending bars. True and false peak detections are expressed as percentages of corresponding peaks detected by the template-matching. Note: no false peak detections were observed at thresholds set to 9, 2, and 1 STD of the noise in steps I, II, and III, respectively. Pie charts below show progress in the true peak detection across the three steps. (**C**) Detectability of signal peaks across arbors of cortical and motor neuron axons. Detectability for the three steps is expressed over axial distances from the putative axon initial segments (AISes). The bottom histogram shows spatial distribution of detected peaks. Data shown in (**B, C**) was obtained from 10 cortical and 10 motor neurons.

inter-branching segments and 23 axon terminals. Axonal activity was detected on 6663 electrodes, occupying an active area of 2.04 mm$^2$. We reconstructed functional morphologies for 50 cortical and 50 motor neurons and analyzed morphological features of their axonal arbors, a total length of 1.04 m and 0.81 m of cortical and spinal axons, respectively.

We found that cortical and motor neurons grew axons comparable in their total lengths. However, ratios between axonal lengths and their corresponding active areas were significantly higher for cortical than spinal axons (**Figure 7B**). The average axonal lengths were 16.73±1.20 and 14.63±0.88 mm for cortical and motor neurons, respectively (p=0.25). The average sizes of active areas were 1.38±0.08 and 1.61±0.08 mm$^2$ for cortical and motor neurons, respectively (p=0.23). The average length-to-area quotients were 11.72±0.32 and 9.49±0.24 mm$^{-1}$ for cortical and motor neurons, respectively (p<10$^{-6}$).

We found cortical axons to develop branching points in more proximal parts of their arbors and to form shorter inter-branching segments as compared to spinal axons (**Figure 7C**). Average axial distances of axonal branching points were 0.27±0.01 and 0.43±0.01 mm for cortical and motor neurons, respectively (p<10$^{-6}$). Average lengths of axonal inter-branching segments were 0.34±0.01 and 0.48±0.01 mm for cortical and motor neurons, respectively (p<10$^{-6}$).

We found significantly more axon terminals in cortical than in spinal axons; however, spinal axons projected their terminals at significantly greater distances as compared to cortical axons (**Figure 7D**). Average numbers of axon terminals were 29.92±1.64 and 17.82±1.03 for cortical and motor neurons, respectively (p<10$^{-6}$). Average axial distances of axon terminals were 1.17±0.02 and 2.41±0.04 mm for cortical and motor neurons, respectively (p<10$^{-6}$).

## Signal amplitude fluctuations during axonal conduction

In addition to exposing the axonal structure, functional morphologies also reveal APs as propagating across axonal arbors (**Figure 8**). They allow observation of AP waveforms at any axonal location and mapping of AP amplitudes across entire arbors. Representative AP waveforms extracted from functional morphologies of cortical and spinal axons are shown in **Figure 8A**, and displayed in **Figure 8—video 1**. We investigated fluctuations in AP amplitudes across functional morphologies of 50 cortical and 50 spinal axons (same neurons as in **Figure 7**). The analysis included 45,232 data points obtained from 1.044 m of cortical axons and 36,286 data points obtained from 0.812 m of spinal axons.

We found that APs recorded from the most proximal parts of axons had much higher amplitudes than APs recorded from more distal axonal locations (**Figure 8A–C**). To further compare AP amplitudes across different cortical and spinal axon domains, we segregated the reconstructed morphologies into

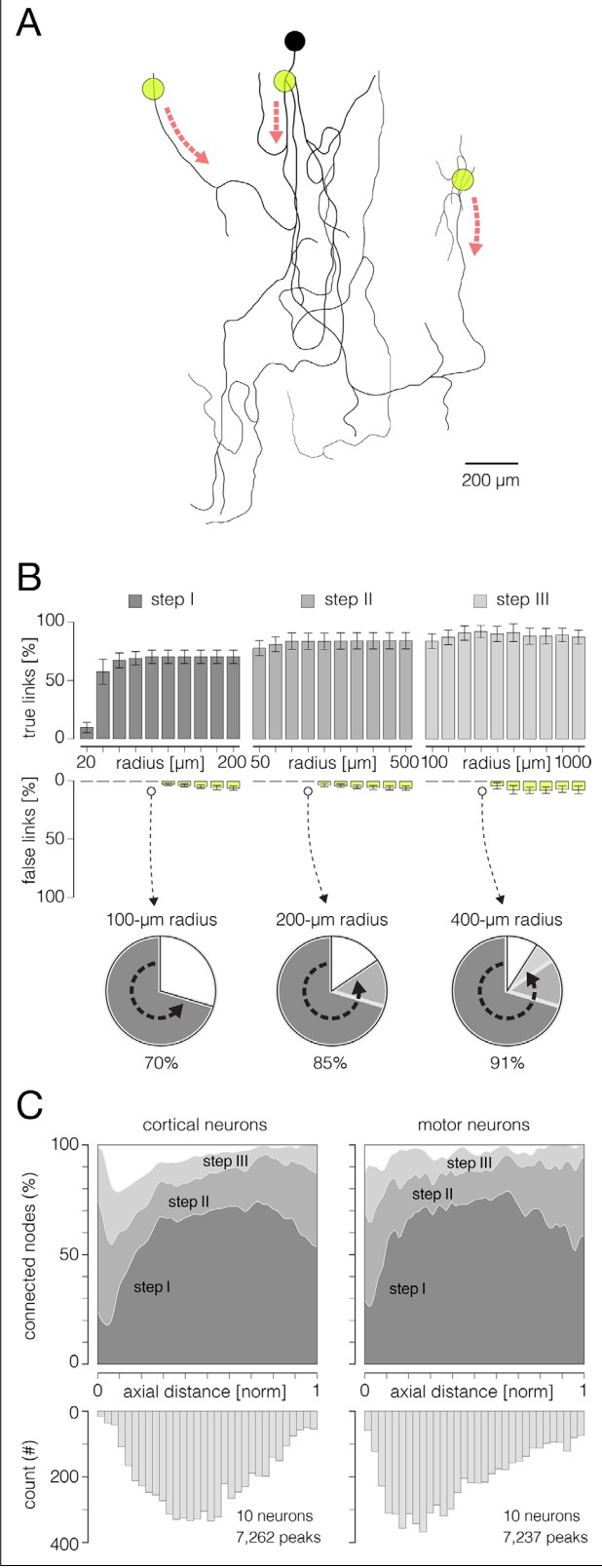

**Figure 6.** Performances of the algorithm for reconstruction of axonal functional morphology. (**A**) Axonal conduction trajectories (black axonal contour) reconstructed from stimulation-triggered neuronal activities; electrical stimulation targeted three different axonal sites (yellow semitransparent circles); red dashed arrows indicate directions of axonal conduction triggered from the corresponding stimulation site. Stimulation-triggered

*Figure 6 continued on next page*

*Figure 6 continued*

action potentials (APs) were reconstructed across an entire array and presented across consecutive movie frames, enabling visual tracking and reconstruction of axonal conduction trajectories (see *Figure 6—videos 1 and 2*). (**B**) Performance of the algorithm for reconstruction of axonal functional morphology. Histograms display percentages of true and false links established between mapped AP peaks using direct, skeletonization-assisted and indirect interconnection (see *Figure 3*). Percentages of true links are exposed through dark-gray, gray, and pale-gray ascending bars (steps I, II and III, respectively); percentages of false links are exposed through yellow descending bars. Percentages of true and false links obtained using different radii for interconnection of the peaks are shown for each of the steps. The percentages are expressed with respect to a total number of links found in the corresponding ground truth trajectory (as shown in A). Note: no false links were established within Euclidean range (radius) of 100, 200, and 400 μm in steps I, II, and III, respectively. Pie charts below show progress in the reconstruction of axonal morphology across the three steps. (**C**) Efficiency of the reconstruction (% of connected nodes) estimated for the three interconnection steps. Efficiencies are shown separately for cortical and motor neuron axons and are expressed over axial distances from the putative axon initial segments (AISes). The bottom histogram shows spatial distribution of connected nodes. Data shown in (**B, C**) was obtained from 10 cortical and 10 motor neurons.

The online version of this article includes the following video and figure supplement(s) for figure 6:

**Figure supplement 1.** Targeted microstimulation of axons using the high-density microelectrode array (HD-MEA) system.

**Figure 6—video 1.** Stimulation-aided reconstruction of axonal conduction trajectories.

https://elifesciences.org/articles/86512/figures#fig6video1

**Figure 6—video 2.** Visually tracked axonal conduction trajectories.

https://elifesciences.org/articles/86512/figures#fig6video2

proximal and distal axons, primary trunks, and axonal trees. Proximal axons pertain to locations found within the first 0.2 mm of axial length; all other locations (axial distance >0.2 mm) were considered as distal (see *Figure 8B*, right). Primary trunk entails locations found in the axonal domain between the soma and the first branching fork, all other locations were assigned to the axonal tree (see *Figure 8C*, right).

APs recorded from proximal axons had significantly higher amplitude in cortical (44.58±4.81 μV) than in motor neurons (19.07±2.53 μV; p<10⁻⁶). On the contrary, APs recorded from distal axons had significantly smaller amplitude in cortical (1.50±0.10 μV) than in motor neurons (3.30±0.38 μV; p<10⁻⁶; *Figure 8B*).

Similarly, APs obtained from primary axonal trunks had significantly higher amplitude in cortical than motor neurons, whereas, APs obtained from axonal trees had significantly lower amplitude in cortical than motor neurons (*Figure 8C*). Average amplitudes of APs recorded from primary axonal trunks were 33.29±4.78 and 15.06±2.78 μV for cortical and motor neurons, respectively (p=0.002). Average amplitudes of APs recorded from axonal trees were 1.25±0.13 and 2.17±0.30 μV for cortical and motor neurons, respectively (p=0.024).

## Temporal dynamics of axonal conduction

Functional morphologies carry information about times at which APs arrived at any axonal site and, as such, provide direct insights into temporal dynamics of axonal conduction (*Figure 9*). Axonal functional morphology reveals time at which axonal conduction is initiated ('initial time') and allows to obtain times at which axonal APs arrive at axon terminals ('arrival time'). It enables extraction of the interval between the earliest and the latest arrival time ('interval of arrivals') and inspection of the total duration of the axonal conduction ('active timespan') (see *Figure 9A*, left). Functional morphologies allow for the inspection of AP conduction dynamics across entire axonal arbors and over individual axonal paths (see *Figure 9A*, right). Finally, structural and temporal parameters from functional morphologies enable computation of conduction velocities for any axonal segment.

We investigated temporal aspects of axonal conduction across functional morphologies of 50 cortical and 50 spinal axons (same neurons as in *Figure 7* and *Figure 8*). The analysis included 45,232 data points obtained from 1.044 m of cortical axons and 36,286 data points obtained from 0.812 m of spinal axons.

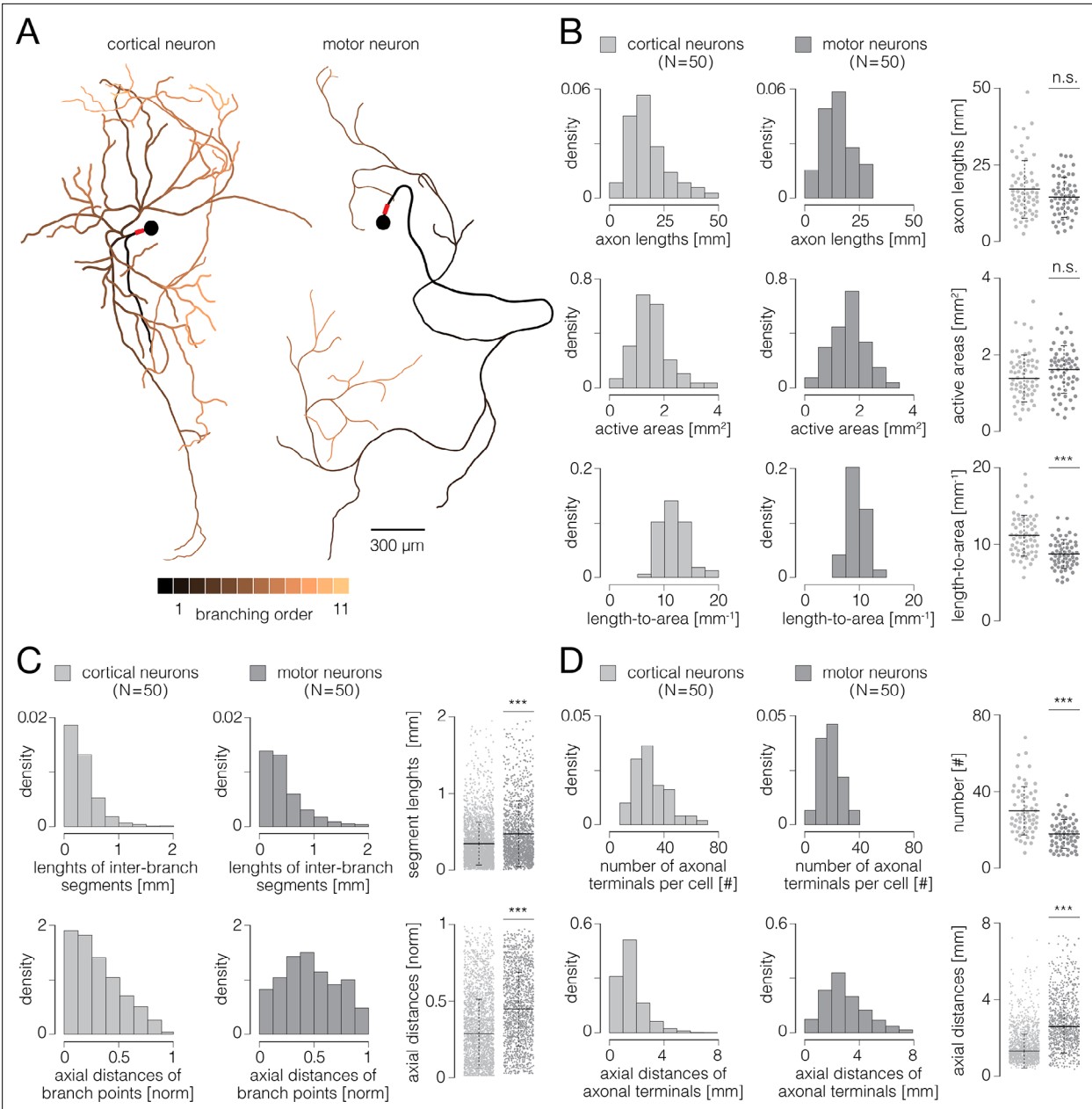

**Figure 7.** Functional morphologies of cortical and spinal axons. (**A**) Functional morphologies of a cortical (left) and motor neuron axons (right) reconstructed based on spontaneous neuronal activities. Branching orders of reconstructed neurites are color-coded; neuronal somas are presented by black-filled circles. Axon initial segments (AIS) are denoted by thick red lines near somas. Functional morphologies of the two neurons are also presented in *Figure 7—video 1*. (**B**) Axonal lengths and active areas. Histograms express density distributions of total axonal lengths, sizes of active areas, and length-to-area quotients. Charts express comparisons between the corresponding values obtained from cortical and motor neurons; horizontal black lines denote mean values, perpendicular black-dashed lines denote standard deviations. (**C**) Lengths of inter-branching segments and axial distances of branching points. (Up) Histograms express density distributions of lengths of inter-branching segments. (Down) Histograms express density distributions of axial distances of branching points (with respect to corresponding AIS). Charts on the right express comparisons between the corresponding values obtained from cortical and motor neurons; horizontal black lines denote mean values, perpendicular black-dashed lines denote standard deviations. (**D**) Quantity and axial distances of axon terminals. (Up) Histograms express density distributions of numbers of axon terminals per cell. (Down) Histograms express density distributions of axial distances of axon terminals (with respect to corresponding AIS). Charts on the right express comparisons between the corresponding values obtained from cortical and motor neurons; horizontal black lines denote mean values, perpendicular black-dashed lines denote standard deviations. Color-code: pale-gray color was used to mark data obtained from cortical neurons; dark-gray color was used to mark data obtained from motor neurons. Data shown in (**B, C, D**) was extracted from functional morphologies of 50 cortical and 50 motor neurons. ***p<0.001.

*Figure 7 continued on next page*

*Figure 7 continued*

The online version of this article includes the following video for figure 7:

**Figure 7—video 1.** Functional morphologies of cortical and spinal axons.

https://elifesciences.org/articles/86512/figures#fig7video1

Cortical neurons had significantly shorter active timespans and higher synchronization of the arrival times as compared to motor neurons (*Figure 9B*). Active timespans were 4.60±0.33 and 7.40±0.32 ms for cortical and motor neurons, respectively ($p<10^{-6}$). Intervals of signal arrivals at axon terminals were 3.44±0.31 and 5.40±0.34 ms for cortical and motor neurons, respectively (p=0.0024). Variance of signal arrival times were 0.95±0.23 and 2.50±0.35 ms$^2$ for cortical and motor neurons, respectively ($p<10^{-4}$).

We next computed axonal conduction velocities using structural and temporal parameters obtained from the functional morphologies. Conduction velocities were calculated from axial distances versus AP propagation times (*Figure 9A*, right), and were computed over 100 µm long axonal chunks stepped by 17.5 µm.

We found cortical axons to exhibit significantly slower and more uniform conduction velocities compared to spinal axons (*Figure 9C*). Average conduction velocities were 0.46±0.01 and 0.56±0.02 m/s for cortical and motor neurons, respectively ($p<10^{-4}$). Variances of conduction velocities were 0.03±0.002 and 0.04±0.003 m/s for cortical and motor neurons, respectively ($p<10^{-4}$).

## Discussion

We developed a method for noninvasive functional imaging of unmyelinated mammalian axons in vitro using the CMOS-based HD-MEA system (*Figure 1*, *Figure 1—videos 1–2*). The method yields an axonal 'functional morphology', comprising multidimensional data derived from extracellular APs recorded during axonal conduction. Functional morphology contains information about axonal conduction trajectories mapped at high spatial and temporal resolution (*Figure 3—videos 1–3* and *Figure 4—videos 1–3*). It allows the reconstruction of axonal 'electrical morphology' at different time-points during the conduction and, at the same time, to expose waveforms of the propagating APs (*Figure 8*, *Figure 8—video 1*).

The presented method has been developed and validated on primary rodent cortical and motor neurons cultured directly over HD-MEA surfaces (*Figure 2—figure supplements 1–3*). The cultures matured within 2 weeks and exhibited spontaneous electrical activities (*Figure 2—figure supplement 6*). They efficiently adhered to HD-MEA surfaces and formed monolayers that varied in local thicknesses and cell densities (*Figure 2—figure supplement 4*). Neurons developed complex axonal arbors and provided a tight interface between axons and microelectrodes. Because axons grew in a 3D manner, their distances from the HD-MEA surfaces varied locally (*Figure 2—figure supplement 5*), but remained within the sensing range of the microelectrodes (*Obien et al., 2019*).

Spontaneous electrical activity enabled us to map individual neurons in the cultures (*Figure 2—figure supplement 6*) and expose the spatiotemporal distribution of APs recorded across their axonal arbors (*Figure 2—figure supplement 7*, *Figure 2—video 3*). Mapping individual neurons was possible, thanks to high-amplitude APs recorded at their AISes (*Bakkum et al., 2019*; *Radivojevic et al., 2016*). The AIS signals broadly surpass the background noise and hence could be easily detected (*Radivojevic et al., 2017*). Because they colocalized with the AIS, local maxima found within activity maps (*Figure 2—figure supplement 6A*) indicate proximal regions of individual neurons in the network (*Bakkum et al., 2019*; *Radivojevic et al., 2016*). Owing to the high-density arrangement of recording microelectrodes (*Figure 2—figure supplement 1*; *Ballini et al., 2014*), spike-sorting enabled discerning mixed signals recorded from neighboring neurons (*Figure 2—figure supplement 6B*) and thereby extracting APs arising from individual neurons (*Figure 2—figure supplement 6C*). Averaging of extracted signals over all electrodes revealed the spatiotemporal distribution of a single neuron's activity. Such signal representation was referred to as an 'axonal electrical image' (*Figure 2—figure supplement 7*, *Figure 2—video 3*) and it carries information about neuronal subcellular elements (*Bakkum et al., 2019*; *Radivojevic et al., 2016*). Thus, the largest and the earliest signals in the footprint arise from the AIS, much smaller and fast propagating signals arise from axons, and slow back-propagating signals arise from the soma and proximal dendrites (*Bakkum et al., 2019*;

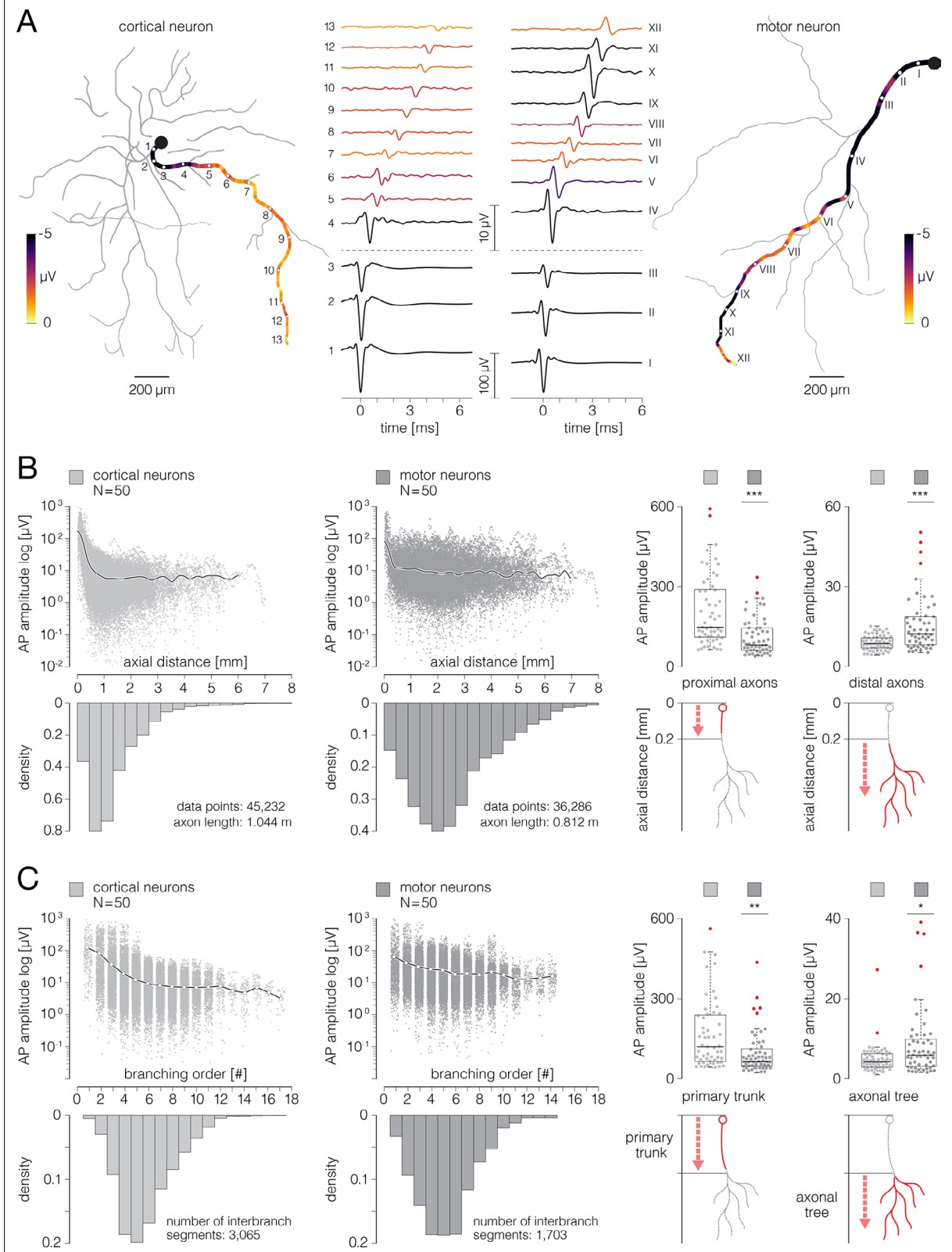

**Figure 8.** Signal amplitude fluctuations during axonal conduction. (**A**) Action potential (AP) waveforms extracted from functional morphologies of cortical and motor neuron axons. (Left) Functional morphology of a single cortical neuron is displayed in gray; average AP amplitudes tracked across selected axonal path are color-coded; waveforms of axonal APs obtained from the denoted locations (numbered 1–13) are shown beside. (Right) Functional morphology of a single motor neuron is displayed in gray; average AP amplitudes tracked across selected axonal path are color-coded;

*Figure 8 continued on next page*

*Figure 8 continued*

waveforms of axonal APs obtained from the denoted locations (numbered I–XII) are shown beside. Note difference in scalebars for AP waveforms obtained from proximal and distal axons. Data from the two neurons are also presented in *Figure 8—video 1*. (**B**) Axonal AP amplitudes versus axial distance from the axon initial segment (AIS). (Left-middle) Amplitudes of axonal APs versus axial distances of their recording sites are plotted for cortical and motor neurons; axial distances are expressed with respect to locations of the corresponding AISes. The black curves represent mean values of amplitudes over all data points. The bottom histograms show density distributions of axial distances from the corresponding AISes. (Right) Mean AP amplitudes obtained from proximal and distal axons are compared between cortical and motor neurons; comparisons are expressed using box plots. The bottom diagrams show criterium for discriminating proximal from distal axonal locations. AP amplitudes averaged over proximal and distal regions of each of the neurons were used in this analysis. (**C**) Axonal AP amplitudes versus axonal branching order. (Left-middle) Amplitudes of axonal AP versus branching order of corresponding axonal segments are plotted for cortical and motor neurons. The white circles (interconnected by black lines) represent mean values of amplitudes computed for discrete branching orders. The bottom histograms show density distributions of branching orders. (Right) Mean signal amplitudes obtained from primary axonal trunks and axonal trees are compared between cortical and motor neurons; comparisons are expressed using box plots. The bottom diagrams show criterium for discriminating primary axonal trunks from axonal trees. Color-code: pale-gray color was used to mark data obtained from cortical neurons; dark-gray color was used to mark data obtained from motor neurons. Data shown in (**B, C**) was extracted from functional morphologies of 50 cortical and 50 motor neurons. **p<0.01; ***p<0.001.

The online version of this article includes the following video for figure 8:

**Figure 8—video 1.** Action potential (AP) waveforms as traced across axonal functional morphologies.
https://elifesciences.org/articles/86512/figures#fig8video1

*Radivojevic et al., 2016*). Thanks to the low-noise CMOS design, the HD-MEA chip allows detection of AP propagation across large portions of entire axonal arbors (*Radivojevic et al., 2017*). The soma and proximal dendrites provide minor contributions to the electrical image, typically masked by larger AIS signals (*Bakkum et al., 2019*). Owing to their low-amplitude extracellular signals, distal dendrites do not seem to be detectable with the HD-MEA system used here.

Adaptive thresholding applied to axonal electrical images enabled us to map extracellular APs propagating across axonal arbors (*Figure 3*, *Figure 3—videos 1–3*). Signal peaks were traced at high spatial and temporal resolution, thanks to the dense arrangement of the electrodes and high-frequency sampling rate (*Ballini et al., 2014*). The thresholding scheme was constructed to detect signals with various amplitudes while minimizing detection errors. Simple planar thresholds, set to 9 STD of the background noise, were used to detect high-amplitude signal peaks (*Figure 3A*, *Figure 3—video 1*). Since these thresholds provided detection high above the level of the background noise, they could be applied across an entire array while yielding no detection errors. They, however, failed to detect low-amplitude signal peaks and only provided fragmentary detection. Instead, spatiotemporal confinement of low-level thresholds, contingent on coordinates of high-amplitude peaks, enabled the mapping of low-amplitude signals (*Figure 3B*, *Figure 3—video 2*). This was possible, thanks to the very nature of axonal conduction, which can be represented using a simplified Markov chain model (*Gagniuc, 2017*). Namely, because signal propagation in unmyelinated axons is continuous, it represents a cascade of successive events in which the probability of each event depends solely on the state attained in the preceding event. Therefore, there was a high probability of detecting signal peaks in the immediate spatiotemporal proximity of previously mapped peaks. While confining the reach of the thresholds to local areas minimized the influence of the background noise, temporal confinement to preceding and following timeframes excluded signals that occurred much earlier and later than the reference peak.

Further relaxation of the threshold confinements enhanced the detection of low-amplitude signals but inevitably increased the risk of detection errors (*Figure 3C*, *Figure 3—video 3*). In principle, our adaptive thresholding scheme followed a greedy algorithm paradigm (*Cormen et al., 2022*), and as such, it is vulnerable to detection errors. Generally, a greedy algorithm builds up a global solution stepwise by making the locally optimal choice at each step. However, erroneous solutions made at earlier steps can proliferate and thereby compromise the global solution (*Gutin et al., 2002*). Therefore, carefully tuned thresholding parameters are key to the optimal detection of signal peaks.

The approach of multiple-moving object tracking enabled us to reconstruct axonal conduction trajectories through iterative associations of the mapped signal peaks (*Figure 4*, *Figure 4—videos 1–3*). The tracking algorithm was designed to estimate inter-frame correspondences between the mapped peaks in a data-driven fashion. It operates within predefined spatiotemporal boundaries, but also directly learns data association criteria from sequences reconstructed along the tracking process. Thanks to strict boundaries, the algorithm discarded trajectories yielding conduction velocities that

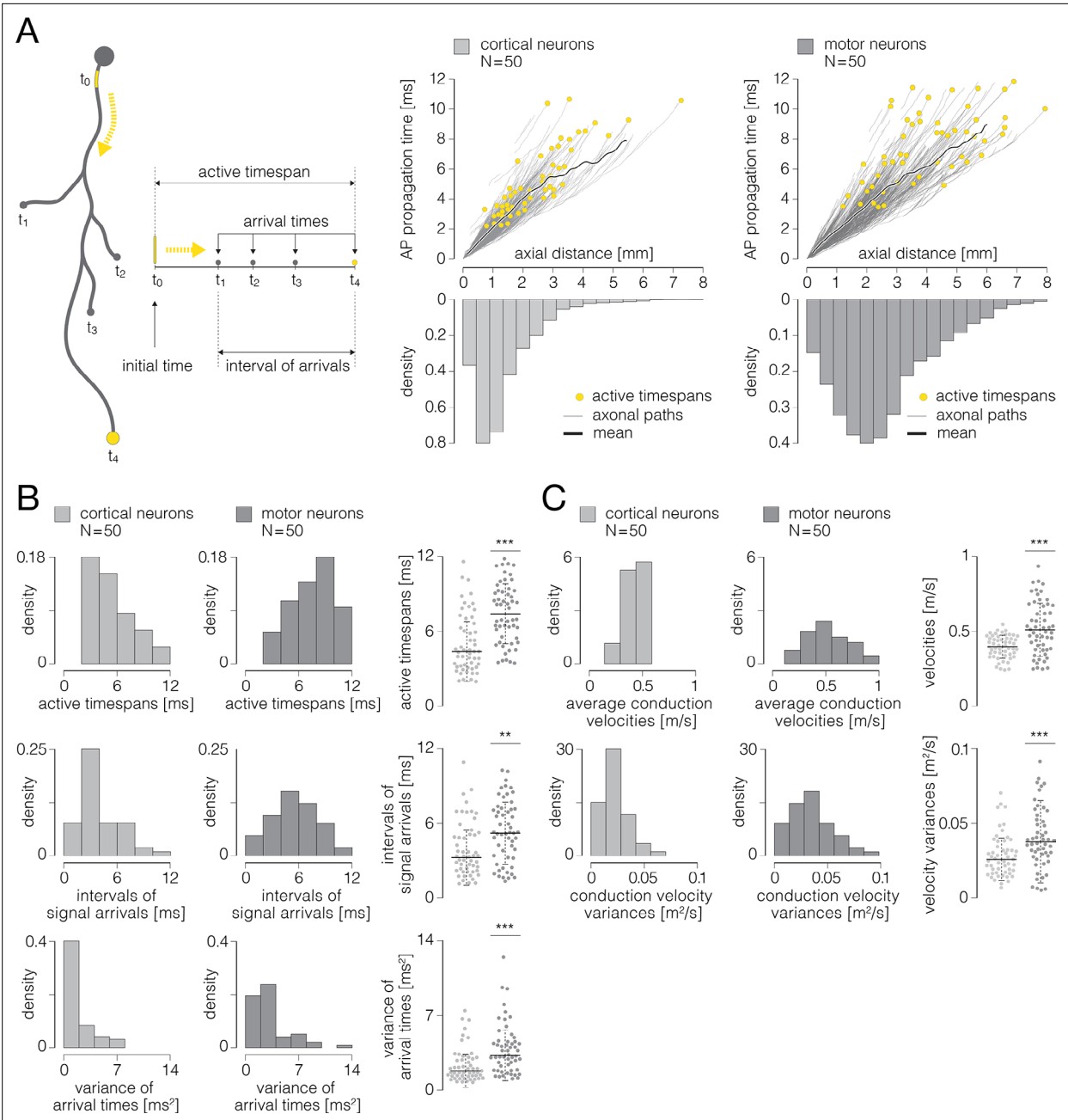

**Figure 9.** Temporal dynamics of axonal conduction. (**A**) (Left) Graphical representation of parameters that were used for the analysis of temporal aspects of axonal conduction. Thick gray contour represents simplified axonal morphology; neuronal somas are presented by gray-filled circles; a yellow line superimposed over a proximal axon represents the axon initial segment (AIS); blebs at the ends of four axonal branches represent axon terminals. Yellow-dashed arrows indicate directionality of axonal action potential (AP) propagation. Temporal parameters: initial time ($t_0$) – time of the initiation of axonal conduction; arrival times ($t_{1-4}$) – times at which axonal APs arrive at axon terminals; interval of arrivals – interval between the earliest and the latest arrival time; active timespan – timespan between initial time and the latest arrival time. (Right) Axonal AP propagation time versus axial distance from the AIS – comparison between cortical (graph on the left) and motor neurons (graph on the right). Yellow circles represent active timespans; gray curves in the background represent AP propagation times observed over individual axonal paths; thick black curves represent mean values of propagation times averaged over all axonal paths. (**B**) Temporal aspects of axonal conduction. (Up) Histograms express density distributions of active timespans for cortical and motor neurons. (Middle) Histograms express density distributions of intervals of arrivals for cortical and motor neurons. (Down) Histograms express density distributions of variances of arrival times for cortical and motor neurons. Charts on the right express comparisons between the corresponding values obtained from cortical and motor neurons; horizontal black lines denote mean values, perpendicular black-dashed lines denote standard deviations. (**C**) Axonal conduction velocities. (Up) Histograms express density distributions of average conduction velocities for cortical and motor neurons; average velocities are computed for individual cells. (Down) Histograms express density distributions of variances of

*Figure 9 continued on next page*

*Figure 9 continued*

conduction velocities for cortical and motor neurons; variances are computed for individual cells. Charts on the right express comparisons between the corresponding values obtained from cortical and motor neurons; horizontal black lines denote mean values, perpendicular black-dashed lines denote standard deviations. Color-code: pale-gray color was used to mark data obtained from cortical neurons; dark-gray color was used to mark data obtained from motor neurons. Data shown was extracted from functional morphologies of 50 cortical and 50 motor neurons. **p<0.01; ***p<0.001.

are unlikely for mammalian axons. While such boundaries ensure the exclusion of slow-propagating dendrites (*Bakkum et al., 2019*), they could also omit super-fast saltatory conduction inherent to myelinated axons (*Debanne, 2004*), which were most likely not present in our preparations. Owing to the data-driven refinement of the association criteria, the algorithm allowed customization of the tracking process for a specific neuron. Such customization provides applicability of the algorithm to axons with different conduction velocities, allowing observation of variations between neurons of the same or different types.

Considering the Markov model of axonal conduction, AP peaks mapped in immediate spatiotemporal proximities can be associated directly using a greedy nearest neighbor matching (*Figure 4A*, *Figure 4—video 1*). Due to the limited reach of the matching caliper, direct interconnection enabled only reconstruction of short fragments of the axonal conduction trajectory. Reconstructed fragments, in return, enabled estimation of the conduction velocity and thereby refining the association criteria. Because conduction velocities vary across a single axon (*Radivojevic et al., 2017*; *Bakkum et al., 2013*), greedy matching was limited to peaks mapped closely over sections of the trajectory with relatively constant conduction velocities. Relaxation of the association criteria could potentially extend the reach of the matching. However, it would inevitably introduce erroneous interconnections between spatially segregated peaks stretched over notably faster sections of the trajectory.

Skeletonization of an axonal electrical image enabled reproduction of conduction paths between distant peaks (*Figure 4B*, *Figure 4—video 2*). Because skeletal remnants reflect the spatiotemporal landscape of the axonal signals, they can optimize the association of mapped peaks graphically, thus sidestepping complex heuristic techniques needed for maintaining computational tractability. In general, topology and structural complexity of skeletal remnants depend on spatiotemporal boundaries within which the axonal signals were skeletonized. Narrow boundaries encompassing two consecutive timeframes captured brief sequences of the axonal conduction and yielded skeletal remnants with fairly simple structure. Simple remnants enabled interconnection of the peaks across consecutive timeframes, however, discontinuous peaks remained beyond their reach.

Skeletonization within wider spatiotemporal boundaries allowed interconnecting peaks across discontinuous timeframes (*Figure 4C*, *Figure 4—video 3*), but yielded more complex remnants, especially in regions where axons branched or intertwined. Complex remnants provided multiple possible solutions for the peak associations, thus aggravating the selection of an optimal conduction path between corresponding peaks. By restricting the skeletonization to three consecutive timeframes and using the refined association criteria, we could select the optimal path in most cases. However, further widening of the skeletonization boundaries increased the number of possible solutions exponentially and inevitably led to a deterioration of tracking performance.

The Bayes optimal template-matching technique was used to validate the algorithm for signal peak detection (*Figure 5*). We chose this technique because it has previously been demonstrated to provide reliable detection of extracellular APs recorded using the HD-MEA system (*Radivojevic et al., 2017*). It enables the detection of axonal APs within single recording trials and microsecond differences in axonal propagation (*Radivojevic et al., 2017*; *Franke et al., 2015*). Bayes optimal template-matching is sensitive enough to discriminate low-amplitude APs from the background noise (*Radivojevic et al., 2017*), and thus can be used to validate our algorithm.

Stimulation-triggered axonal activities allowed us to trace axonal conduction trajectories accurately and thus provided the ground truth for validation of our tracking algorithm (*Figure 6*, *Figure 6—videos 1 and 2*). We have previously shown that both spontaneous and stimulation-triggered neuronal activities recorded by HD-MEAs can be used to outline axonal conduction trajectories (*Bullmann et al., 2019*; *Radivojevic et al., 2016*). Moreover, we demonstrated a congruence between the reconstructed trajectories and actual axonal morphology revealed optically (*Radivojevic et al., 2017*; *Bakkum et al., 2013*; *Bullmann et al., 2019*; *Radivojevic et al., 2016*).

Previous research on HD-MEA systems provided methods for the electrical imaging of axons (*Radivojevic et al., 2017*; *Bakkum et al., 2013*; *Müller et al., 2015*; *Bullmann et al., 2019*; *Radivojevic et al., 2016*) but did not provide algorithms for the reconstruction of axonal morphology. Therefore, optical imaging of axonal morphology or manual tracking of AP propagation was necessary to assign the recorded signals to their respective axonal locations (*Radivojevic et al., 2017*; *Bakkum et al., 2013*; *Bullmann et al., 2019*; *Radivojevic et al., 2016*). Recent studies have proposed algorithms for deducing axonal arbors and propagation velocity based on extracellular APs (*Yuan et al., 2020*; *Buccino et al., 2022*). However, these algorithms have solely been validated on simulated morphologies, and no experimental validation has been done to assess their performances (*Buccino et al., 2022*). The present algorithm enables precise reconstruction of axonal electrical morphology and has been validated on experimental data obtained from cortical and motor neurons (see *Figures 5 and 6*). The algorithm's performances were investigated, and optimal parameters for accurate reconstruction were identified. The algorithm's adaptive functionality enables it to adjust and optimize its parameters in a data-driven fashion, which allows for customizing the tracking process for a specific neuron.

We investigated functional morphologies of cortical and spinal axons and found significant differences between their structures, AP amplitudes, and conduction dynamics (*Figures 7–9*), which potentially reflect biological hallmarks of different neuronal subtypes. Although comparable in overall length, cortical and spinal axons exhibited substantially different functional morphologies. Cortical axons had a more complex branching pattern than spinal axons, but spinal axons projected their terminals to much longer distances than cortical axons. The amplitude of APs recorded from spinal axons was higher than those recorded from cortical axons. However, when comparing signals recorded from the AIS, cortical neurons displayed significantly higher amplitude than motor neurons. Cortical neurons had shorter active timespans and more synchronized arrival times than motor neurons, while cortical axons had slower and more uniform conduction velocities than spinal axons.

The structural complexity of cortical axons was reflected in their extensive branching, followed by the abundance of the axon terminals that were projected locally and distally (*Figure 7*, *Figure 7—video 1*). Such morphological features of cortical axons can be attributed to their role in providing synaptic connectivity within local neuronal assemblies as well as across large neuronal networks, thus enabling higher-order operations performed in the brain (*Buzsáki, 2010*). Spinal motor neurons, however, have evolved much simpler morphology to provide fundamentally different biological roles. Motor neurons innervate and precisely control muscles in the periphery outside the CNS and are, for that purpose, equipped with the longest known axons in the body (*Stifani, 2014*). Our results obtained in in vitro conditions are consistent with such a notion. They indicate that spinal axons indeed have much simpler morphologies when compared to cortical axons, yet still retain relatively long total lengths (*Figure 7*, *Figure 7—video 1*). We found that spinal axons had moderately branched arbors, but projected their axon terminals at great distances, thus resembling morphological adaptations observed in vivo. Thanks to their extensively branched axons, cortical neurons enabled the delivery of APs to numerous axon terminals, surprisingly synchronously and within short timespans. On the contrary, spinal axons with less branches at their disposal required considerably more time to forward their APs to axon terminals in a less synchronized fashion (*Figure 9B*). In this study, the assessment of the structural complexity of cortical and spinal axons relies entirely on their functional morphologies. It would be valuable for future studies to compare the morphological features obtained through electrical and optical imaging.

We found that extracellular APs recorded from proximal axons near the putative AIS exposed disproportionately larger amplitudes than signals obtained from other axonal parts (*Figure 8*, *Figure 8—video 1*). It has been previously shown that the AIS is the dominant contributor to the neuron's extracellular electrical landscape (*Bakkum et al., 2019*), which can be attributed to the high densities of voltage-gated ion channels expressed in the AIS (*Kole et al., 2008*; *Hu et al., 2009*; *Lorincz and Nusser, 2010*). Interestingly, we found that APs obtained from proximal axons had significantly higher amplitudes in cortical than in motor neurons (*Figure 8B and C*). The relatively low AP amplitudes at the AIS in motor neurons could result from the potentially disrupted organization of the AIS in unmyelinated motor neurons. Such disruption pertains to altered distribution and combination of voltage-gated ion channels as well as incomplete segregation of the AIS, para-AIS, and juxtapara-AIS compartments inherent to myelinated spinal axons (*Duflocq et al., 2011*). The diameter, length,

and positioning of the AIS in relation to recording electrodes may also substantially impact shaping the extracellular AP profile (*Bakkum et al., 2019*; *Kumar et al., 2022*), which in turn can influence the AP amplitude. We found that signals obtained from distal axons and axonal trees had significantly lower amplitudes in cortical than motor neurons (*Figure 8B and C*). This difference could be explained by the fact that spinal axons have a considerably larger diameter than cortical axons (*Saliani et al., 2017*). As such, they engage a greater number of voltage-gated ion channels and create stronger transmembrane currents during an AP electrogenesis (*Stein and Pearson, 1971*). However, a comprehensive modeling study that incorporates biophysical, structural, and spatial parameters is necessary to explain the observed differences in AP amplitudes between cortical and spinal axons. Large diameters of spinal axons could also explain their significantly faster conduction velocities compared to cortical axons (*Figure 9C*). In general, axonal diameter and the presence of myelin sheaths are crucial factors that control conduction velocity in mammalian axons (*Debanne et al., 2011*; *Waxman, 1980*). On the one hand, conduction velocity of unmyelinated axons is proportional to the square root of the axon diameter (*Hodgkin and Huxley, 1952*). On the other hand, myelination provides a fundamentally different mechanism of AP propagation known as 'saltatory' conduction (*Huxley and Stämpeli, 1949*). In this case, myelin sheaths spatially restrict the distribution of voltage-gated ion channels to nodes of Ranvier and thus impose discontinuous AP propagation up to 100-fold faster than propagation in unmyelinated axons (*Debanne et al., 2011*; *Saliani et al., 2017*). The culture protocols used in this study do not support the formation of myelin sheaths around axons. Therefore, it is unlikely that complete myelination occurred in our cultures. Namely, the myelination of primary neurons in vitro requires either supplement that supports well-balanced development of both neurons and myelin-producing cells (*Pang et al., 2012*) or, in the case of motor neurons, co-culture with a feeder layer of Schwann cells (*Hyung et al., 2015*). In both cases, the timely addition of specific neurotrophic factors is required to initiate the myelination program. The culture media used in this study were not specifically designed to support the growth of myelin-producing cells, and no myelination-promoting factors were included in the media. The protocols did not involve co-culturing motor neurons with Schwann cells that would provide myelination. Moreover, motor neurons were grown in a culture medium containing NT3, a neurotrophic factor known to significantly inhibit myelination in vitro (*Chan et al., 2001*). Finally, we did not detect any saltatory high-speed signal conduction in any recorded neurons, suggesting that myelin sheaths were not formed around the axons. We, however, cannot exclude potential cases where axons were partially myelinated. Moreover, incomplete myelinization or discontinuous distribution of voltage-gated ion channels could explain relatively large variances of conduction velocities in spinal axons (*Figure 9C*).

Generally, standard deviations of the data presented in *Figures 7–9* could result from morphological and biophysical differences between various neuronal subtypes that may have existed in our cultures. Thus, for example, developed cortical cultures comprise excitatory glutamatergic and inhibitory GABAergic neurons which are known to differ in size and morphological complexities (*Björklund et al., 2010*). Even greater diversity of neuronal subtypes can be found among motor neurons, including alpha, beta, and gamma motor neurons that are structurally specialized to innervate different muscle fiber types in the body (*Stifani, 2014*). However, another question is to what extent such specializations can be recapitulated in vitro and whether electrophysiological parameters observed in cultures allow for discriminating between neuronal subtypes.

The presented method for functional imaging of cortical and spinal axons provides direct insights into the biophysical properties of axonal conduction. It allows investigation of interdependences between axonal function and structure, noninvasively and over extended periods. Functional morphologies contain data obtained from large portions of axonal arbors and, as such, allow studying the fidelity of signal conduction in different axonal elements, including conduction failures at branching points (*Debanne, 2004*) and signal attenuation in tiny axon terminals (*Faisal and Laughlin, 2007*). They also allow the inspection of activity-dependent modulation of extracellular AP waveforms (*Radivojevic et al., 2017*; *Lewandowska et al., 2016*) and analysis of consequent changes in times at which modulated signals arrive at axon terminals (*Radivojevic et al., 2017*). Owing to its noninvasive nature, the method enables the study of structural and functional changes in axons over long periods. Thus, for example, comparative analysis of functional morphologies obtained at different timepoints allows investigation of biophysical changes during axonal outgrowth and pathfinding – developmental processes that guide axons toward specific targets and enable wiring within neuronal networks (*Myers*

*et al., 2011*; *Uesaka et al., 2005*). Complemented with techniques for electrical microstimulation (see *Figure 6—figure supplement 1*), the method can be used to study activity-dependent axonal plasticity, including structural alteration such as the positional shift of the AIS (*Grubb and Burrone, 2010*; *Kumar et al., 2022*) as well as functional adaptations that involve changes in axonal conduction velocities (*Bakkum et al., 2013*). Because the HD-MEA system allows observing electrical activities across entire neuronal networks (see *Figure 2—figure supplement 6*), the method can serve as a complementary tool for studying the functional interplay between network dynamics and plastic adaptations in axons (*Grubb and Burrone, 2010*; *Kuba et al., 2010*).

In summary, the presented method is designed for specific tissue-on-a-chip platforms and may serve as an attractive tool for pharmacological and preclinical studies of neurodegenerative diseases (*Ronchi et al., 2021*). The key potential of the method lies in its ability to gain multilevel knowledge of neuronal functionality, which may serve as a valuable asset in drug discovery and safety pharmacology. The method holds great potential for developing disease-specific bioassays, especially considering the advent of human induced pluripotent stem cell technology that allows for utilizing patient-derived neurons.

# Methods

**Key resources table**

| Reagent type (species) or resource | Designation | Source or reference | Identifiers | Additional information |
|---|---|---|---|---|
| Strain, strain background (*Rattus norvegicus*) | Sprague Dawley rat | Charles River Laboratories | Strain Code: 400 | – |
| Cell line (Rattus *norvegicus*) | Cortical; Spinal | This paper | – | Primary |
| Antibody | Anti-Map2 (Chicken Polyclonal) | Abcam | ab5392 | IHC (1:1000) |
| Antibody | Anti-β-III tubulin (Rabbit Polyclonal) | Abcam | ab18207 | IHC (1:500) |
| Antibody | Anti-mCherry (Mouse Monoclonal) | Abcam | ab125096 | IHC (1:500) |
| Antibody | Alexa Fluor 488 (Goat Anti-Chicken Polyclonal) | Abcam | ab150173 | IHC (1:200) |
| Antibody | Alexa Fluor 488 (Donkey Anti-Rabbit Polyclonal) | Abcam | ab150073 | IHC (1:200) |
| Antibody | Alexa Fluor 594 (Goat Anti-Mouse Polyclonal) | Abcam | ab150116 | 1 IHC (:200) |
| Recombinant DNA reagent | pAAV-hSyn-mCherry | Addgene | Plasmid # 114472 | – |
| Peptide, recombinant protein | Human recombinant laminin | BioLamina | LN211 | – |
| Software, algorithm | Spyking Circus | *Yger et al., 2018* | Spyking Circus | Version 1.0.1 |

## Animal use

All experimental protocols were approved by the Uppsala Animal Ethical Committee under animal license C97/15 and follow the guidelines of the Swedish Legislation on Animal Experimentation (Animal Welfare Act SFS 2009:303) and the European Communities Council Directive (2010/63/EU).

## HD-MEA and signal processing

A CMOS-based HD-MEA system (MaxWell Biosystems AG) was used for extracellular neuronal recording and stimulation (*Figure 2—figure supplement 1*). The array comprises 26,400 platinum microelectrodes (9.3×5.45 μm$^2$) packed within an area of 3.85×2.10 mm$^2$, providing a density of 3150 electrodes per mm$^2$ (17.5 μm center-to-center pitch). Owing to a flexible switch matrix technology, up to 1024 readout and/or stimulation channels could be routed to the desired electrodes and reconfigured within a few milliseconds. On-chip circuitry was used to amplify (0–80 dB programmable gain), filter (high pass: 0.3–100 Hz, low pass: 3.5–14 kHz), and digitize (8-bit, 20 kHz) the neuronal signals. Digitized signals were sent to a field-programmable gate array board and further streamed to a

host PC for real-time visualization and data storage. Recorded signals were up-sampled to 200 kHz following the Whitaker-Shannon interpolation formula. Python 3.7 and Matlab R2020a were used for data analysis and to design experimental protocols.

## Cortical cultures

We used a culturing protocol for the long-term maintenance of neural cultures (*Hales et al., 2010*: *Figure 2—figure supplement 2*). Minor adaptations were introduced to the protocol to constrain culture growth to the sensing area of the array and maintain optimal growth media conditions during long-term experimentation (*Radivojevic et al., 2016*). Cortices from embryonic day 18 Sprague Dawley rat were dissociated enzymatically in trypsin supplemented with 0.25% EDTA (Thermo Fisher Scientific) and physically by trituration. The remaining cell aggregates and debris were filtered using 40 µm cell strainer (Corning). For cell adhesion, a layer of 0.05% polyethyleneimine (Sigma-Aldrich) in borate buffer (Chemie Brunschwig), followed by a layer of 0.02 mg/ml human recombinant laminin (BioLamina) in DPBS containing $Ca^{2+}$ and $Mg^{2+}$ (Thermo Fisher Scientific) was deposited on the electrode array. To constrain culture growth to the electrode array, a cell drop containing ~50,000 cells and covering ~4 $mm^2$ was seeded in the center of the array. The plating media were changed to growth media after 24 hr and regularly changed every 6 days. Plating media consisted of Neurobasal, supplemented with 10% horse serum (HyClone), 0.5 mM GlutaMAX, and 2% B27 (Thermo Fisher Scientific). The growth media consisted of DMEM, supplemented with 10% horse serum (HyClone), 0.5 mM GlutaMAX, and 1 mM sodium pyruvate (Thermo Fisher Scientific). Cultures were maintained inside an incubator under controlled environmental conditions (36°C and 5% $CO_2$). The culturing chambers were sealed with an ~1 mm layer of light mineral oil (Sigma-Aldrich) floating above the growth medium. The sealing provided selective permeability to gases, such as $O_2$ and $CO_2$, and prevented evaporation and consequent changes in the growth media's osmolarity during long-term experiments.

## Motor neuron cultures

Protocol for growing motor neuron cultures on HD-MEA surface is illustrated in *Figure 2—figure supplement 3*. Spinal cords isolated from embryonic day 14 Sprague Dawley rat were dissociated enzymatically in trypsin supplemented with 0.25% EDTA (Thermo Fisher Scientific) and physically by trituration. The remaining cell aggregates and debris were filtered out using 40 µm cell strainer (Corning). A density gradient medium composed of 15% OptiPrep (Sigma-Aldrich) in Leibovitz's L-15 medium (Thermo Fisher Scientific) was used to fractionate motor neurons. For cell adhesion, a layer of 0.05% polyethyleneimine (Sigma-Aldrich) in borate buffer (Chemie Brunschwig), followed by a layer of 0.02 mg/ml human recombinant laminin (BioLamina) in DPBS containing $Ca^{2+}$ and $Mg^{2+}$ (Thermo Fisher Scientific) was deposited on the electrode array. To constrain culture growth to the electrode array, a cell drop containing ~50,000 cells and covering ~4 $mm^2$ was seeded in the center of the array. The growth media consisted of Neurobasal supplemented with 20% MyoTonic differentiation medium (Cook MyoSite), 1% fetal bovine serum, 2% horse serum, 2% B27, 1% antibiotic-antimycotic, 0.2 mM gentamicin, 0.7 mM L-glutamine, 1.5 mM sodium pyruvate (Thermo Fisher Scientific), 0.8 nM brain-derived neurotrophic factor, 0.2 nM glial cell line-derived growth factor, 0.2 nM ciliary neurotrophic factor, 1.5 nM neurotrophin 3 (NT3), 1.4 nM neurotrophin 4 (NT4, R&D Systems), and 1.3 nM human insulin-like growth factor 1 (hIGF-1, Thermo Fisher Scientific). The growth media were changed after 24 hr and further regularly changed every 4 days. Cultures were maintained inside an incubator under controlled environmental conditions (36°C and 5% $CO_2$). The culturing chambers were sealed with an ~1 mm layer of light mineral oil (Sigma-Aldrich) floating above the growth medium.

## Immunocytochemistry

Neuronal cultures were fixed in 4% paraformaldehyde (Thermo Fisher Scientific) in PBS (Sigma-Aldrich) at pH 7.4 for 15 min at room temperature, washed twice with ice-cold PBS, permeabilized with 0.25% Triton X-100 (Sigma-Aldrich) in PBS for 10 min and washed three times in PBS. Fixed cultures were exposed to phosphate-buffered saline with Tween 20 (1% bovine serum albumin and 0.1% Tween 20 in PBS; Sigma-Aldrich) for 30 min to prevent unspecific binding of antibodies. The primary antibodies Anti-Map2 (Abcam, ab5392), Anti-β-III tubulin (Abcam, ab18207), and Anti-mCherry (Abcam, ab125096), diluted in phosphate buffered saline with Tween 20 to ratios of 1:1000, 1:500, and 1:500 respectively, were added and left overnight at 4°C on a shaker. Cultures were washed three times in

PBS for 5 min each time on the shaker. The secondary antibodies Alexa Fluor 488 (Abcam, ab150173), Alexa Fluor 488 (Abcam, ab150073), and Alexa Fluor 594 (Abcam, ab150116) diluted to ratio of 1:200 in PBS with 1% BSA were added and left for 60 min in the dark at room temperature. Samples were washed three times in PBS for 5 min in the dark and then stored at 4°C. Cultures were immunostained with anti-β-III tubulin (*Figure 2—figure supplements 4 and 5*, *Figure 1—videos 1 and 2*), anti-Map2 (*Figure 2—figure supplements 6 and 7*, *Figure 2—videos 1–3*), and a combination of anti-mCherry and anti-β-III tubulin antibodies (*Figure 2—figure supplement 5*).

### Live imaging

Live-cell visualization of neurons was performed by transfection using pAAV-hSyn-mCherry plasmid from Karl Deisseroth (Addgene plasmid # 114472) and Lipofectamine 3000 (Thermo Fisher Scientific), jetPRIME (Polyplus) or TurboFect (Thermo Fisher Scientific) transfection reagent following the manufacturer's protocols. Micrographs of cortical and motor neurons transfected using Lipofectamine 3000 are shown in *Figure 2—figure supplement 5*.

### Microscopy and 3D image reconstruction

A Nikon Eclipse LVDIA-N microscope, Nikon DS-Fi2 camera, and the Nikon NIS-Elements imaging software were used to produce micrographs. Epifluorescence microscopy was used to collect Z-stack image series from immunostained cultures shown in *Figure 2—figure supplements 4 and 5* and *Figure 2—videos 1 and 2*. ImageJ and Matlab custom designed codes were used to create 3D surface plots (*Figure 2—figure supplement 4*) and to reconstruct 3D neuronal morphologies (*Figure 2—figure supplement 5*) based on the intensities of immunofluorescent signals.

### Network-wide activity mapping

Spontaneous extracellular APs were sampled across the entire microelectrode array by sequential scanning over 28 recording configurations. Up to 1024 randomly selected electrodes recorded neuronal activity in each configuration for 2 min. The average voltage traces, recorded by each electrode, were used to reconstruct the network-wide activity map (*Figure 2—figure supplement 6A*). Since the largest extracellular APs occur near the AIS, and APs arising from axonal arbors have much smaller amplitude, regions with high-amplitude APs in the activity maps indicated the locations of the AISes (*Radivojevic et al., 2017*; *Bakkum et al., 2019*; *Radivojevic et al., 2016*).

### Electrical identification of individual neurons

Simultaneous access to signals arising from the AIS region enabled us to reconstruct the spatiotemporal distribution of extracellular APs, referred to as the 'electrical footprint' (*Figure 2—figure supplement 6C*). To obtain these signals, we used high-density recording configurations covering the AIS locations revealed in the activity maps (*Figure 2—figure supplement 6A*). In each configuration, blocks of 13×13 electrodes were connected to readout channels to sample neuronal activity for 2 min. Recorded signals were sorted by using the Spyking-Circus algorithm (*Yger et al., 2018*), and electrical footprints were reconstructed by using custom-designed Matlab code. Because the first-occurring (initial) trace found in the electrical footprint colocalizes with the neuron's AIS, it was used to trigger the averaging of voltage traces recorded across an entire axonal arbor (see below).

### Electrical imaging of axonal arbors

Array-wide averaging of voltage traces, synchronized with the initial trace (recorded from the AIS), reveals the spatiotemporal distribution of extracellular APs across an entire axonal arbor (*Figure 2—figure supplement 7* and *Figure 2—video 3*). The first step in obtaining these data was selecting 9 electrodes that were closest to the putative AIS. We next designed multiple recording configurations covering the entire array – in each configuration, 9 of the 1024 readout channels were routed to the 9 preselected electrodes, and remaining available channels were routed to randomly selected electrodes. Each configuration was used to sample neuronal activity during 2 min. Signals recorded by the 9 preselected electrodes in each configuration were sorted using the Spyking-Circus algorithm (*Yger et al., 2018*). Timestamps of the sorted signals were used to trigger the averaging of voltage traces recorded across all other electrodes in the array. The spatiotemporal distribution of averaged signals was reconstructed using a custom-designed Matlab code.

## Electrical stimulation of individual neurons in the network

Stimulation performances of the HD-MEA system allow for stimulating any neuron in the culture with subcellular spatial precision (*Radivojevic et al., 2017*; *Bakkum et al., 2013*; *Radivojevic et al., 2016*; *Ronchi et al., 2019*). Key concepts of the stimulation protocols that were used in this study are presented in *Figure 6—figure supplement 1*. Extracellular electrical stimulations targeted at predefined axonal locations allow to experimentally control the direction of the axonal conduction (*Radivojevic et al., 2016*; *Figure 6—figure supplement 1A*). Electrical stimulation directed at the AIS was used to elicit orthodromic neuronal activation (*Figure 6 – figure supplement 1A*, left). Stimulation directed at distal axons was used to elicit antidromic neuronal activation (*Figure 6 – figure supplement 1A*, right). All stimulation protocols utilized balanced positive-first biphasic voltage pulses, with phase durations of 200 µs, because of their proven effectiveness in electrical stimulation (*Wagenaar et al., 2004*). Stimuli were applied to one electrode at a time at a frequency of 4 Hz. Neuronal responses to orthodromic stimulations were estimated by observing APs recorded from distal axons (*Figure 6 – figure supplement 1A*, left). Responses to antidromic stimulation were estimated by observing APs recorded from proximal axons and the AIS (*Figure 6 – figure supplement 1A*, right). To find optimal parameters for stimulation of individual neurons in the network, we used approaches thoroughly described in our previous study (*Radivojevic et al., 2016*). In brief, we applied neuron-wide stimulation over a range of voltages to reveal sites with the lowest activation threshold (*Figure 6—figure supplement 1B*). Stimulation was applied at 4 Hz for voltages from ±10 to ±300 mV, with steps of ±10 mV. Each stimulation voltage was applied 60 times per site, and activation thresholds were defined as the minimum voltage to trigger an AP in 100% of the trials. To get more detailed excitability profiles, the most sensitive sites were then stimulated with voltages stepped by ±1 mV (*Figure 6—figure supplement 1C*).

## Statistical analysis

All quantitative data presented in *Figures 7–9* are expressed as mean ± SEM. Numbers of biological replicates (N) used for analyses presented in *Figures 5–9* are denoted in the figures and are also stated in their legends. We used non-parametric tests for comparing distributions of parameters presented in *Figures 7–9*, since normal distribution of the underlying data could not be determined unequivocally. The two-sided Mann-Whitney U-test was applied and a p-value <0.05 was considered significant. Logarithmic scaling of graphs shown in *Figure 8B and C* was used to display a wide range numerical data in a compact way.

## Acknowledgements

We thank Olga Netsyk, Bryndis Birnir, Tanel Punga, and Evgenii Bogatikov for help with experiments; Markus de Ruijter and Stefan Engblom for help with software developments; Marta K Lewandowska and Yu-Fang Huang for critical discussions. This study was supported by Swedish Research Council grant 2016-02184, Swedish Research Council grant 2014-02048, Swedish Research Council grant 2014-07603, Swedish Research Council grant 2020-02040, Göran Gustafsson foundation for medical research, Hjärnfonden (the Swedish Brain Foundation), Olle Engkvist Byggmästare grant 197-0235, Margarethahemmets foundation and Bissen Brainwalk Foundation. The funders had no role in study design, data collection and analysis, decision to publish, or preparation of the manuscript.

## Additional information

### Funding

| Funder | Grant reference number | Author |
| --- | --- | --- |
| Vetenskapsrådet | 2016-02184 | Anna Rostedt Punga |
| Vetenskapsrådet | 2014-02048 | Anna Rostedt Punga |
| Vetenskapsrådet | 2014-07603 | Anna Rostedt Punga |
| Vetenskapsrådet | 2020-02040 | Anna Rostedt Punga |

| Funder | Grant reference number | Author |
|---|---|---|
| Göran Gustafssons Stiftelse för Naturvetenskaplig och Medicinsk Forskning | | Anna Rostedt Punga |
| Hjärnfonden | | Anna Rostedt Punga |
| Olle Engkvists Stiftelse | 197-0235 | Anna Rostedt Punga |

The funders had no role in study design, data collection and interpretation, or the decision to submit the work for publication.

### Author contributions

Milos Radivojevic, Conceptualization, Resources, Data curation, Software, Formal analysis, Validation, Investigation, Visualization, Methodology, Writing - original draft, Project administration, Writing - review and editing; Anna Rostedt Punga, Resources, Funding acquisition, Writing - review and editing

### Author ORCIDs

Milos Radivojevic (ID) http://orcid.org/0000-0001-8828-9475

### Ethics

All experimental protocols were approved by the Uppsala Animal Ethical Committee under animal license C97/15 and follow the guidelines of the Swedish Legislation on Animal Experimentation (Animal Welfare Act SFS 2009:303) and the European Communities Council Directive (2010/63/EU).

### Decision letter and Author response

Decision letter https://doi.org/10.7554/eLife.86512.sa1
Author response https://doi.org/10.7554/eLife.86512.sa2

## Additional files

### Supplementary files

- MDAR checklist

### Data availability

Datasets generated for this study, entitled 'Electrical imaging of cortical and spinal axons using high-density microelectrode arrays', are stored in Dryad.

The following dataset was generated:

| Author(s) | Year | Dataset title | Dataset URL | Database and Identifier |
|---|---|---|---|---|
| Radivojevic M, Rostedt Punga A | 2023 | Electrical imaging of cortical and spinal axons using high-density microelectrode arrays | https://doi.org/10.5061/dryad.gxd2547r1 | Dryad Digital Repository, 10.5061/dryad.gxd2547r1 |

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
