## [Editor Report]

This work provides fundamental new insight into fine axonal morphologies based solely on extracellular action potential recordings. They provide compelling evidence of fine resolution in mapping functional connections between neurons. The work may have broad use in neurobiology, bioengineering, stem cell biology, as well as tissue engineering in functional characterization.

---

## [Decision Letter]

**Decision letter after peer review:**

Thank you for submitting your article "Functional imaging of conduction dynamics in cortical and spinal axons" for consideration by *eLife*. Your article has been reviewed by 2 peer reviewers, one of whom is a member of our Board of Reviewing Editors, and the evaluation has been overseen by Panayiota Poirazi as the Senior Editor.

The editors and the reviewers find this to be a very interesting and compelling paper reporting a method for analyzing the features of action potential conduction in cortical and spinal neurons in vitro using high-density CMOS micro-electrode arrays. The authors provide compelling evidence of fine resolution in mapping functional connections between neurons via very fine axons. The advantage of the approach is that it provides a label-free electrical visualization of axon conduction trajectories as well as the ability to access the AP waveforms. The work may have broad use in neurobiology, bioengineering, stem cell biology as well as tissue engineering in functional characterization.

Essential revisions:

1. Summary: please conclude on the differences found in cortical and spinal axons.

2. Highlight the performances of the present algorithm compared to previous studies by the Hierlemann lab (i.e., Bakkum et al., 2013, etc).

3. What is the evidence for the lack of myelination in the cultured neurons?

4. Figure 8: what could be the explanation for the marked increase in spike amplitude in segment X of the spinal axon? A higher density of sodium channels? If this is the case, perhaps the authors could provide immunostaining of Nav channels or AnkB/G.

5. Add additional immunostaining to confirm the morphological findings from the array.

*Reviewer #1 (Recommendations for the authors):*

I would recommend expanding the repertoire of standard imaging modalities via e.g. immunostaining (some already shown in Figure S5) as well as possibly TEM to confirm morphologically some of the findings from the MEA recordings that led to the reconstruction of the morphological maps.

*Reviewer #2 (Recommendations for the authors):*

1. Summary: please conclude on the differences found in cortical and spinal axons.

2. Highlight the performances of the present algorithm compared to previous studies by the Hierlemann lab (i.e., Bakkum et al., 2013, etc…).

3. What is the evidence for the lack of myelination in the cultured neurons?

4. Figure 8: what could be the explanation for the marked increase in spike amplitude in segment X of the spinal axon? A higher density of sodium channels? If this is the case, perhaps the authors could provide immunostaining of Nav channels or AnkB/G.

---

## [Author Response]

Essential revisions:Reviewer #1 (Recommendations for the authors):I would recommend expanding the repertoire of standard imaging modalities via e.g. immunostaining (some already shown in Figure S5) as well as possibly TEM to confirm morphologically some of the findings from the MEA recordings that led to the reconstruction of the morphological maps.

Figure S5 is now Figure 2 —figure supplement 5.

We thank reviewer 1 for this very constructive recommendation. Incorporating the analysis of axonal morphologies obtained optically would be a valuable addition to our paper, providing a complementary perspective to our findings derived from electrical recordings. However, obtaining morphological features and metrics corresponding to those provided by functional morphology requires considerably more effort than expanding the repertoire of standard imaging modalities. Unfortunately, it is unfeasible for this study.

Specifically, functional morphologies reconstructed from electrophysiological signals provide metrics for total axonal length, lengths of inter-branching segments, and number and axial distances of axonal branching points and terminals. Extracting these features from optically obtained data requires labeling individual neurons in the culture and accurate reconstruction of their axonal arbors. Because standard immunostaining labels all axons in a culture, it does not allow for distinguishing between the axonal arbors of individual neurons. While transfecting cultures with plasmids encoding fluorescent reporters can provide sparse labeling of individual neurons (see Figure 2 —figure supplement 5 – left), the axons of labeled neurons still intersect extensively, presenting a challenge for accurate reconstruction of their axonal arbors. Reconstructed axonal morphologies shown in Figure 2 —figure supplement 5 – right can be used to estimate the axonal distance from the HD-MEA surface accurately. However, these reconstructions are only partial, capturing fragments of axonal arbors, and are not precise enough to extract morphological features yielded by functional morphologies. Optical data from high-resolution epifluorescence or immunogold SEM imaging, capturing entire axonal arbors, could enable accurate reconstruction of axonal morphologies. However, it is unfeasible to gain access to the necessary specialized equipment and obtain this data within a reasonable timeframe. Finally, conducting comparative analyses between morphological features obtained through electrical and optical imaging is better done in an independent follow-up study.

In response to this comment, we edited the Discussion section on page 17 / lines 427-429 to clarify our standpoint.

“In this study, the assessment of the structural complexity of cortical and spinal axons relies entirely on their functional morphologies. It would be valuable for future studies to compare the morphological features obtained through electrical and optical imaging.”

Reviewer #2 (Recommendations for the authors):1. Summary: please conclude on the differences found in cortical and spinal axons.

In response to this comment, we edited the Discussion section on page 16 / lines 406-412.

“Although comparable in overall length, cortical and spinal axons exhibited substantially different functional morphologies. Cortical axons had a more complex branching pattern than spinal axons, but spinal axons projected their terminals to much longer distances than cortical axons. The amplitude of APs recorded from spinal axons was higher than those recorded from cortical axons. However, when comparing signals recorded from the AIS, cortical neurons displayed significantly higher amplitude than motor neurons. Cortical neurons had shorter active timespans and more synchronized arrival times than motor neurons, while cortical axons had slower and more uniform conduction velocities than spinal axons.”

2. Highlight the performances of the present algorithm compared to previous studies by the Hierlemann lab (i.e., Bakkum et al., 2013, etc…).

In response to this comment, we edited the Discussion section on pages 15 and 16 / lines 394-403 and added citations 38 and 39 to contextualize the performances of the present algorithm with respect to previous relevant studies.

“Previous research on HD MEA systems provided methods for the electrical imaging of axons but did not provide algorithms for the reconstruction of axonal morphology. Therefore, optical imaging of axonal morphology or manual tracking of AP propagation was necessary to assign the recorded signals to their respective axonal locations. Recent studies have proposed algorithms for deducing axonal arbors and propagation velocity based on extracellular APs. However, these algorithms have solely been validated on simulated morphologies, and no experimental validation has been done to assess their performances. The present algorithm enables precise reconstruction of axonal electrical morphology and has been validated on experimental data obtained from cortical and motor neurons (see Figures 5 and 6). The algorithm's performances were investigated, and optimal parameters for accurate reconstruction were identified. The algorithm's adaptive functionality enables it to adjust and optimize its parameters in a data-driven fashion, which allows for customizing the tracking process for a specific neuron.”

3. What is the evidence for the lack of myelination in the cultured neurons?

The culture protocols used in this study do not support the growth of oligodendrocytes or Schwann cells needed for the formation of myelin sheaths. Additionally, we did not detect any saltatory high-speed signal conduction in any recorded neurons, suggesting that myelin sheaths were not formed around the axons. We, however, cannot exclude potential cases where axons were partially myelinated. As we have no further evidence to support or exclude this possibility, we cannot make any exclusive claims.

In response to this question, we edited the Discussion section on pages 17 and 18 / lines 452-463 and added citations 49-51 to clarify this point.

“The culture protocols used in this study do not support the formation of myelin sheaths around axons. Therefore, it is unlikely that complete myelination occurred in our cultures. Namely, the myelination of primary neurons in vitro requires supplements that support the well-balanced development of neurons and myelin-producing cells or, in the case of motor neurons, co-culture with a feeder layer of Schwann cells. In both cases, the timely addition of specific neurotrophic factors is required to initiate the myelination program. The culture media used in this study were not specifically designed to support the growth of myelin-producing cells, and no myelination-promoting factors were included in the media. The protocols did not involve co-culturing motor neurons with Schwann cells that would provide myelination. Moreover, motor neurons were grown in a culture medium containing NT3, a neurotrophic factor known to significantly inhibit myelination in vitro. Finally, we did not detect any saltatory high-speed signal conduction in any recorded neurons, suggesting that myelin sheaths were not formed around the axons. We, however, cannot exclude potential cases where axons were partially myelinated.”

4. Figure 8: what could be the explanation for the marked increase in spike amplitude in segment X of the spinal axon? A higher density of sodium channels? If this is the case, perhaps the authors could provide immunostaining of Nav channels or AnkB/G.

The lower AP amplitudes observed at the AIS in motor neurons may be attributed to the potential disruption in the organization of the AIS in unmyelinated motor neurons. This disruption could involve alterations in the distribution and combination of voltage-gated ion channels and incomplete segregation of the AIS, para-AIS, and juxtapara-AIS compartments typically found in myelinated spinal axons. The diameter, length, and positioning of the AIS in relation to recording electrodes may also substantially impact shaping the extracellular AP profile, which in turn can influence the AP amplitude. Therefore, not solely the local densities of voltage-gated ion channels can explain the lower AP amplitudes observed at the AIS in motor neurons. To accurately address this question, it would be necessary to conduct comprehensive modeling studies that consider all of these parameters, which was not possible within the scope of this study.

In response to this comment, we edited the Discussion section on page 17 / lines 438-440 and 444-445, and added citation 45 to clarify this point.

“The diameter, length, and positioning of the AIS in relation to recording electrodes may also have a substantial impact on shaping the extracellular AP profile, which in turn can influence the AP amplitude.”

“However, a comprehensive modeling study incorporating biophysical, structural, and spatial parameters is necessary to explain the observed differences in AP amplitudes between cortical and spinal axons.”